# Precision Neuro-Oncology in Glioblastoma: AI-Guided CRISPR Editing and Real-Time Multi-Omics for Genomic Brain Surgery

**DOI:** 10.3390/ijms26157364

**Published:** 2025-07-30

**Authors:** Matei Șerban, Corneliu Toader, Răzvan-Adrian Covache-Busuioc

**Affiliations:** 1Puls Med Association, 051885 Bucharest, Romania; mateiserban@innbn.com (M.Ș.); razvancovache@innbn.com (R.-A.C.-B.); 2Department of Neurosurgery, “Carol Davila” University of Medicine and Pharmacy, 050474 Bucharest, Romania; 3Department of Vascular Neurosurgery, National Institute of Neurology and Neurovascular Diseases, 077160 Bucharest, Romania

**Keywords:** precision neurosurgery, genomic medicine, CRISPR-Cas9, glioblastoma, IDH mutation, epilepsy surgery, multi-omics, single-cell sequencing, molecular imaging, neurodegenerative diseases

## Abstract

Precision neurosurgery is rapidly evolving as a medical specialty by merging genomic medicine, multi-omics technologies, and artificial intelligence (AI) technology, while at the same time, society is shifting away from the traditional, anatomic model of care to consider a more precise, molecular model of care. The general purpose of this review is to contemporaneously reflect on how these advances will impact neurosurgical care by providing us with more precise diagnostic and treatment pathways. We hope to provide a relevant review of the recent advances in genomics and multi-omics in the context of clinical practice and highlight their transformational opportunities in the existing models of care, where improved molecular insights can support improvements in clinical care. More specifically, we will highlight how genomic profiling, CRISPR-Cas9, and multi-omics platforms (genomics, transcriptomics, proteomics, and metabolomics) are increasing our understanding of central nervous system (CNS) disorders. Achievements obtained with transformational technologies such as single-cell RNA sequencing and intraoperative mass spectrometry are exemplary of the molecular diagnostic possibilities in real-time molecular diagnostics to enable a more directed approach in surgical options. We will also explore how identifying specific biomarkers (e.g., IDH mutations and MGMT promoter methylation) became a tipping point in the care of glioblastoma and allowed for the establishment of a new taxonomy of tumors that became applicable for surgeons, where a change in practice enjoined a different surgical resection approach and subsequently stratified the adjuvant therapies undertaken after surgery. Furthermore, we reflect on how the novel genomic characterization of mutations like DEPDC5 and SCN1A transformed the pre-surgery selection of surgical candidates for refractory epilepsy when conventional imaging did not define an epileptogenic zone, thus reducing resective surgery occurring in clinical practice. While we are atop the crest of an exciting wave of advances, we recognize that we also must be diligent about the challenges we must navigate to implement genomic medicine in neurosurgery—including ethical and technical challenges that could arise when genomic mutation-based therapies require the concurrent application of multi-omics data collection to be realized in practice for the benefit of patients, as well as the constraints from the blood–brain barrier. The primary challenges also relate to the possible gene privacy implications around genomic medicine and equitable access to technology-based alternative practice disrupting interventions. We hope the contribution from this review will not just be situational consolidation and integration of knowledge but also a stimulus for new lines of research and clinical practice. We also hope to stimulate mindful discussions about future possibilities for conscientious and sustainable progress in our evolution toward a genomic model of precision neurosurgery. In the spirit of providing a critical perspective, we hope that we are also adding to the larger opportunity to embed molecular precision into neuroscience care, striving to promote better practice and better outcomes for patients in a global sense.

## 1. Introduction

### 1.1. Background

Precision medicine has altered our grasp of diseases based on distinct genetic, molecular, and environmental characteristics of patients. Driven by the availability of genomic technologies such as next-generation sequencing (NGS) and single-cell transcriptomics, it has expanded our understanding of the mechanistic underpinnings of diseases and has noteworthy implications for oncology and cardiology [1]. Neurosurgery, which is precise by definition, is now in a position to undergo the same transformation as genomic medicine in terms of incorporation into neurosurgical practice [2,3].

Historically, neurosurgical expertise has relied on macroscopic tools, including sophisticated imaging and anatomical mapping, to provide guidance for the surgical intervention. Although these neuroimaging techniques have improved the precision of surgical interventions, they are limited in terms of addressing the molecular and genetic complexities underlying diseases, for example, glioblastomas, refractory epilepsy, or neurodegenerative disorders. In the past, the histological classification of brain tumors failed to incorporate molecular heterogeneity, which drove the prognosis and therapeutic response or failure in the wake of histological classification [4]. In the same vein, the use of conventional imaging often fails to detect non-lesional epilepsy and early neurodegenerative changes [5]. This calls for the implementation of a neurosurgical approach that employs molecular information related to neurological diseases.

Genomic medicine aims to fill this gap by elucidating the molecular architecture of neurological diseases and facilitating the onset of neurological disease-specific interventions guided by the patient’s individual biology. Tools including CRISPR-Cas9, transcriptomic profiling, and intraoperative genomic sequencing are impacting our understanding of how a neurosurgeon gains diagnostic clarity, undertakes surgical planning, and executes interventions [6]. Each of these technologies further establishes precision medicine by informing neurosurgical and therapeutic approaches to illness. One example of the adoption of interventions informed by genome sequencing is intraoperative sequencing, where the surgeon can dynamically adapt the approach with respect to the molecular microenvironment of tumors, preserving critical brain regions while exerting therapeutic actions [7].

This review seeks to provide a framework to understand how genomic medicine is impacting practice in neurosurgery by categorizing advances into domains of diagnostic assessment, surgical precision, and personalized treatment. Additionally, advances in neuro-genomics are converging with other significant advancements beyond genomics, such as artificial intelligence (AI) and molecular imaging, and these will help accelerate the future of precision neurosurgery. This review intends to provide insights into the ethical, regulatory, and practical considerations relevant to these developments in neurosurgical practice and proposes a way forward in terms of embedding the result of these transformative technologies in routine practice for neurosurgery. The topic of neuro-genomic medicine is now at an inflection point for neurosurgery, whereby interventions are guided not only by anatomical visualization but also by the molecular blueprint of the etiology of each condition. Ultimately, a neurosurgeon will benefit from genomic understanding, thereby surpassing traditional limitations to disease and reframing the traditional moribund condition in some conditions within the central nervous system (CNS). The intention of this paper is to provide clarity in this trajectory and delineate the future of precision in neurosurgical practice.

We sought to construct a review that is both readable and clinically useful to aid clarity and understanding. We also aimed to respect a translational continuum which starts with molecular genetic alterations, navigates through pathophysiologic processes, and ends with descriptions of innovative surgical and genomic treatments. It would be beneficial to visualize the association between the gene, the disease, the pathophysiology, and the clinical interventions, especially for those linking molecular neuroscience to the neurosurgical clinic; to this end, we employed summary tables and specific figures that catalogue gene–disease associations, functional deficits, and therapeutic strategies (which included CRISPR™ strategies).

Further, we aimed to decrease fragmentation by associating concepts and making signaling connections in various sections. Given the vast and rapidly progressing fields involved, we tried to keep the text consistent, useful, and clinically relevant. We believe this review outlines a simple and meaningful framework for what precision medicine is starting to look like in the future of neurosurgical care.

### 1.2. Importance in Neurosurgery

How would neurosurgery be if it could go beyond what can be seen and integrated into what can be understood through a microscope? Genomic medicine is making this a reality and is changing the practice of neurosurgery by revealing molecular mechanisms relating to neurological diseases. Neurosurgery was once predicated on anatomy and histological features which are visible through a microscope; now, there is a genetic and molecular landscape of disease processes upon which interventions can be executed based on a patient’s individual biology.

Brain tumors are a perfect example. Patients historically have been classified by histological features as assessed under a microscope. This method was not only variable, but it could not encapsulate the biological diversity that exists within such lesions. The molecular signatures of brain tumors have also been defined through genomic profiling, including, but not limited to, IDH mutations, TERT promoter alterations, and 1p/19q co-deletions [8,9]. Biologically defining tumor subtypes allows not only for a better representation of the mutant biology of the tumor type but also for better predictions of patient outcomes and responses to therapy. Perhaps the most innovative process has been the ability to modulate surgical plans in real time using intraoperative genomic sequencing of tumors so that surgeons can now better decide the limits of tumor resection while avoiding critical structures in the brain or curing brain tumors, which was impossible 10 years previously [10].

This potential is transformational to neurodegenerative diseases as well. Neurodegenerative diseases such as Alzheimer’s disease (AD) and Parkinson’s disease (PD), which we previously treated as clinical syndromes, are now regarded as molecularly complicated processes [11,12]. Genetic components such as the APOE ε4 allele, which is responsible for Alzheimer’s disease, and mutations in LRRK2, which are responsible for Parkinson’s disease, are now facilitating the development of individualized approaches for neurosurgical or pharmacological treatments [13]. In the case of Parkinson’s disease, genetic components may promote deep brain stimulation (DBS) to facilitate the precision of electrode placement or stimulation modes by taking into consideration the individual genetic background [14]. This is simply an example of using biological data from genetics and molecular biology to enhance efficacy of treatment and reduce potential side effects [15].

In the field of epilepsy, genomic medicine is rapidly transforming the identification and selection process of candidates for surgery. For decades, patients were identified as candidates for epilepsy surgery by strictly imaging and electroclinical correlations. However, with molecular advances in genetics, we may now identify pathologic mutations in genes like SCN1A or KCNQ2 for specific types of epilepsy syndromes [16,17]. The identification of this molecular pathology will improve patient selection and also broaden the spectrum of patients into the realm of neurosurgical treatments that were previously considered untreatable [18]. Additionally, new and innovative technologies in molecular genomics like transcriptomics profiling are emerging and being tested in the operating room to provide real time, intraoperative molecular maps [19].

Perhaps the most exciting area of discovery is the intraoperative, real-time potential of genomic editing. Tools such as CRISPR-Cas9 were developed for molecular research and are currently being evaluated for their potential as intraoperative tools to edit pathogenic mutations [20]. If this is successful, we may view this as a paradigm shift for patients with diseases like gliomas or in inheritable monogenic epilepsies, as treatment may no longer be limited to simply symptomatic care and potentially allow for clinical alteration of the disease course, providing new hope for cure [21].

Shifting the focus of clinical medicine from the macro anatomical aspects of the disease and disorder to molecular blueprints of disease is adding another level of precision to neurosurgery. Bidirectionally, this may help the surgeon, patient, and family to better understand brain tumors, neurodegenerative diseases, and epilepsy and also provide better outcomes and redefine the treatable space in patients with CNS diseases.

The progression toward molecular precision has unfolded over many decades based on many steps, with each producing additional layers of clarity and precision. Early neurosurgery was predicated simply on the surface anatomy and palpation; subsequent advances transitioned the field multiple times. The development of CT and MRI allowed for lesion localization and planning, while neuronavigation and stereotactic instrumentations ushered in the possibilities of relocation and location-based accuracy. Slowly, as surgical imaging improved, the question evolved from Landau’s notion of subject anatomy to subject biology, and this study has evolved to real time sequencing, multi-omics, and CRISPR based interventions that represent additional steps, serving us not only where we operate, but increasingly interrogating the suitability of the intervention, the quantity, and recipient. Artificial Intelligence builds on this foundation; the magnitude of data outcomes across modalities and carriers allows it to provide alternatives that lead to informed decision making that is predictive, individualized, and biologically accurate throughout an entire patient journey. This grand arc from generality in anatomy to precision in molecular being and computational understanding is a transformation of neurosurgical or, if you prefer, of surgical identity, all of which is no longer anatomical or procedural, but integrative, adaptive, and grounded in a constant reiteration of each patient’s biography, which is revelatory of their unique biological life cycle.

### 1.3. Objectives of the Review

In developing this paper, our aim was to provide a focused and conceptually coherent synthesis of the emerging genomic and computational technologies and to discuss how these are beginning to change neurological thought and neurosurgical practice. While we acknowledge the breadth of innovation in this field, we decided to focus on specific conditions: glioblastoma, drug-resistant epilepsy, and major neurodegenerative diseases including Alzheimer disease and ALS, that are best exemplified the intersection of molecular profiling, intraoperative technologies, and individualized treatment planning. Each example was intended to exemplify the larger trends observed across precision neurosurgery. Rather than provide a systematic catalogue, we aimed to highlight the translational pathways that are currently influencing surgical decision-making, rather than those that may emerge from CRISPR technologies, AI-based predictive tools, or off-the-shelf tools that incorporate multiple dimensions of information, aka, multi-omics. We hope that this selective but strategic approach provides clarity regarding the scope of the review and potentially useful insights for clinicians and researchers working in different subspecialties in neurosurgery.

This review begins a dialogue on how the merging and converging of genomic medicine with neurosurgery is continuously changing “can” and “must” treat CNS conditions. It aims to:Describe how genomic profiling will change our conceptualization and management of patients with brain tumors, neurodegenerative diseases, and epilepsy while also facilitating our uptake of that molecular knowledge into an actionable evidence-based standard of care.Describe innovative technologies such as intraoperative genomic sequencing, transcriptomics, and CRISPR-mediated gene editing to improve precision or treatment effects in surgery.Discuss how molecular imaging modalities (e.g., PET, molecular MRI) might work with genomic knowledge to improve preoperative planning and intraoperative decision making.Explore the use of artificial intelligence to merge genomic, radiological, and clinical datasets to develop predictive models that can lead to the individualization of care in neurosurgery.Address the ethical and societal ramifications of this research agenda such as genetic privacy, equitable access, and regulatory concerns regarding the use of genomic technology in surgical practice.Offer ideas for next steps to push this research agenda toward the future of precision neurosurgery; specifically, to stimulate creative and cross-discipline collaboration for the purpose of expediting the ambition of integration of genomic medicine into the clinical neurosurgical workflow.

By synthesizing these individual endeavors, this review intends to establish an overarching vision of what the future of precision neurosurgery could look like. In doing so, it emphasizes the shift from static, universal ways of addressing a disease to increasingly dynamic, individualized, personalized, and contextualized approaches to care that focus on the unique molecular signature of diseases which make up the illness that brought the patient to the surgical context.

In an emerging area of scientific inquiry and clinical practice, from brain tumors to genetically mediated epilepsies, the biological underpinning of clinical or anatomic outcomes exists at the intersection of genomics and innovative technologies—a place where science intersects humanity to improve outcomes. In selecting the technologies discussed in this review, we wanted to emphasize those technologies that, although at differing stages of clinical maturity, signal a paradigm shift in our understanding of and interactions with the brain in the surgical context. Though tools like optogenetics and nanotechnologies are not yet in use as a routine part of the neurosurgical workflow, they provide valuable examples of cellular-scale modulation and targeted delivery, including real-time molecular interventions. While our aim is to invoke these emergent approaches not as speculative at present but rather as conceptually significant continuations and demonstrations of a logical trajectory toward biologically-precise and minimally-invasive neurosurgical care, our own examples of already available technologies are a window into the horizon we are heading toward, which we hope can pave the way for future interdisciplinary collaborations.

This review intended to examine and synthesize current concepts, findings, and applications at the intersection of neurosurgery, genomic medicine, and technological innovation. Relevant literature was identified by broadly and continually surveying published material, primarily derived from PubMed, Google Scholar, and references of landmark articles. Peer-reviewed articles, translational advances from current research, and literature that considered a conceptual or applied clinical basis for the ideas presented were all emphasized. The chosen literature includes laboratory findings, clinical research, and emerging technologies to indicate the direction of travel. Our focus was not on every possible advancement but rather on identifying representative trends and providing meaningful observations that represent the developing nature of precision neurosurgery.

## 2. Evolution of Neurosurgery in the Genomic Era

The discipline of neurosurgery, which was once relegated to anatomical visualization and histopathological means of assessment, has entered a period of unparalleled transition under the umbrella of genomics in medicine. This transition is more than just a technical evolution, but a vertically enhanced evolution, with regard to the surgical toolbox, with a depth of molecular thinking. The purpose of this section is to follow these evolving pathways from traditional perspectives into precision neurosurgery, to offer the reader a panoramic view of how neurosurgery is redefining its boundaries and maximizing its possibilities.

### 2.1. Traditional Neurosurgical Approaches

Traditionally, neurosurgery was dependent on structural imaging and anatomical knowledge in an additive way, as with the interventions or surgical approaches of Harvey Cushing and his ice maker, through his paradigm shifting technique for brain tumor resection, or Walter Dandy and his introduction of ventriculography to neurosurgery, or the development of other methods for CSF anatomy. The history of the practice has been a continuous endeavor for better technology to improve accuracy and safety, a relentless effort to find technological solutions. The introduction of imaging modalities that included computed tomography (CT) and magnetic resonance imaging (MRI) changed the landscape of our ability to visualize and assess relevant intracranial lesions and their anatomical relationship to adjacent anatomical structures; these imaging modalities were, and remain, ubiquitous [22,23,24].

We have also been able to apply functional neurosurgical imaging modalities that include positron emission tomography (PET) and functional MRI (fMRI), which have opened the gateway for additional opportunities in the field of neurosurgery, with respect to providing contextual information regarding degrading regions of the surface of the brain that are objectively tasked with functionality or metabolic processes. The increased utility of these technologies affords the opportunity for the surgeon to make informed decisions in terms of analyses of electrode positions or plotting with respect to the repercussions of operating on temporally functioned areas, e.g., motor cortex, language etc. [25]. There are many situations when preop fMRI mapping of the frontal motor and language areas have become a routine presurgical process, including quantifications or fMRIs, as described in various published studies on motor skills and language [26].

Despite improvements in imaging and functional imaging, traditional neurosurgical approaches remain limited, as they do not necessarily engage the contextual perplexities of the molecular portion of the processes of neurological diseases [27]. Histopathological classification of brain tumors, for example, assigned biologically diverse entities in broad categories, such as glioblastoma multiforme (GBM), which did not account for the genetic heterogeneity that drives tumor behavior, treatment resistance, and outcomes in patients [28]. Epilepsy surgery also focused on localizing possible benefit zones solely based on electroencephalogram (EEG) findings and imaging, dismissing the role of genetic mutations in seizure activity [29].

Neurodegenerative diseases presented an even greater challenge for traditional neurosurgical paradigms. With disorders such as Alzheimer’s and Parkinson’s disease, there are no distinct (surgically targetable) lesions to address, leaving these diseases nearly untreatable in a traditional surgical sense. This represented an important gap in neurosurgery, specifically, the inability to address the underpinning of disease on a molecular and/or genetic level, which also has an effect on pathogenesis, disease course, and response to therapy [30].

### 2.2. Shift Toward Precision Neurosurgery

Adding genomic medicine to the field of neurosurgery is a true paradigm shift, moving from anatomical specific interventions to interventions of molecular specifics and precision. This transition is not only attributable to advancements in molecular technology and milestones in genomic studies but is also due to our understanding that neurological diseases are inherently heterogeneous in nature on a molecular level. The addition of genomic medicine to the field of neurosurgery is similar to its re-definition of surgical treatment in fields such as oncology and cardiology; however, it also presents the field with unique challenges and opportunities [31]. Genomic profiling has transformed cancer care in oncology by classifying tumors into molecular subtypes. This has paved the way for precision therapies including EGFR inhibitors for lung cancer and BRAF-targeted treatments for melanoma. In cardiology, the identification of genetic risk factors like those underlying familial hypercholesterolemia has led to more evidence-based prevention strategies and individual pharmacological treatment alternatives [32].

The field of neurosurgery, on the other hand, functions within the unique context of the CNS. The complexities of the blood-brain barrier (BBB) and the limited natural regenerative capacity of the brain raise challenges which are not as present in other contexts. In contrast to peripheral tumors, brain pathologies frequently require courses of action that treat the disease but also allow for the preservation of critical functions [33]. For example, the identification of IDH mutations has profoundly affected the classification and prognosis of gliomas; however, turning these molecular insights into surgical decisions during operative cases requires an unprecedented level of synthesis of genomic and surgical workflows [34,35].

Moreover, neurosurgery regularly deals with disease processes such as epilepsy and Parkinson’s, which can have substantial genetic influences on phenotypic expression. While the hallmark of oncology is slowing tumor growth, neurosurgical measures often seek a balance between disease control and preserving function. An example of this type of precision medicine being utilized in neurosurgery is the notion of tailoring DBS to the various subtypes of Parkinson’s disease. This notion of precision allows the surgeon to produce improved patient outcomes by using molecular profiles and overlapping stimulation parameters to guide stimulation [36].

These examples emphasize the importance of considering the need for neurosurgery to adapt the approaches of genomic technologies to the demands of medical care in neurosurgery. Transcriptomic and epigenomic profiling show promise through their incorporation of the dynamic biological processes that drive neurologic disease in the CNS. Aligning molecular profiling with the surgical and therapeutic precision of care, the field of neurosurgery is leading the way for meaningful and personalized interventions that are transforming what is possible for some of the most challenging conditions of neurology [37].

The molecular reclassification of brain tumors is an example of a genomic sequencing approach as a means to revolutionize the practice of neurosurgery. The discovery of isocitrate dehydrogenase (IDH) mutations in gliomas made it possible to identify different metabolic and clinical traits that could not have been established previously [38,39]. For example, IDH-mutant gliomas are noted for their less aggressive, slower growth and improved response to therapy when compared to an IDH wild-type glioma. The ability to recognize the presence of IDH mutations led to their incorporation into the 2016 World Health Organization (WHO) classification of CNS tumors, along with other important attributes such as 1p/19q co-deletions and MGMT promoter methylation annotations [40,41].

The molecular pathway in relation to IDH mutations is illustrated in Figure 1, which shows how mutant IDH enzymes catalyze the inappropriate conversion of isocitrate to the oncomolecule 2-hydroxyglutatrate. This aberrant metabolism alters cellular metabolism and leads to widespread and significant epigenetic changes, contributing to DNA damage and chromatin paucity with the IDH-mutant gliomas. This pathway further solidifies the relevance of IDH mutations not only as diagnostic biomarkers, but also as therapeutic targets representing molecular precision.

Molecular profiling has provided important new information in meningiomas, the most common primary brain tumors. Recurrent mutations in the TRAF7, KLF4, and NF2 genes are associated with distinct subtypes that differ in their recurrence risk and response to treatment. This is enabling the development of targeted therapies for higher risk or recurrent tumors that cannot completely be managed by surgical intervention [42,43].

The discovery of genomic information has revolutionized epilepsy surgery through the identification of the genetic basis of many epilepsy syndromes. Mutations in genes such as SCN1A, KCNQ2, and DEPDC5 are associated with specific epileptic phenotypes and allow for molecular stratification of patients [44,45]. For example, DEPDC5 mutations associated with focal cortical dysplasia can now provide genetic and imaging data to prioritize patients for resective surgery. In addition to genomic work, transcriptomic studies have provided unique patterns of gene expression in epileptogenic tissue architectures that can provide new potential biomarkers intraoperatively [46].

Technological advances, such as laser interstitial thermal therapy (LITT), can also be enhanced by molecular consideration to provide a higher degree of accuracy in treatment plans. LITT is a minimally invasive technique that is being adopted in circumstances where resective surgery would not be possible, e.g., in the peri-eloquent cortex or deep brain structures [47].

Neurodegenerative diseases can benefit from genomic work and can redefine offerings for neurosurgical interventions. For example, in Alzheimer’s disease, genome-wide association studies (GWASs) provide more than 40 genetic loci associated with the risk of disease progression, including variability of APOE ε4 and TREM2. These advancements are informing the creation of neurosurgical delivery systems for anti-amyloid and anti-tau treatments (i.e., CED and focused ultrasound) that circumvent the BBB in order to address the targeted areas [48,49].

In PD, the genetic stratification of LRRK2, GBA, and SNCA mutations has reshaped the practice of DBS, as these genetic data are utilized for personalized electrode/treatment positioning based on genetic modifiable variables, ultimately leading to improvements in motor and non-motor variables. Next, intraoperative technologies will continue to push precision medicine into neurosurgery. Fluorescence-guided surgery with (e.g., 5-ALA) is already a gold standard in glioblastoma resection, identifying and resecting tumor margins with great accuracy. Intraoperative technologies, using mass-spectrometry-based techniques (e.g., DESI) enable real-time characterization of resected tissues and guide decisions on how far to resect while also considering future therapy [50,51].

Similarly, NGS has been adapted for intraoperative utility to provide actionable data once surgery has started. This has been remarkably impactful in glioblastomas, where the genomic characterization further informs intraoperative decisions on resection and treatment [52].

Finally, AI and machine learning (ML) are emerging powerhouses in precision neurosurgery. Specifically, by combining radiological and genomic data, radiogenomic models are being built to predict outcomes such as tumor recurrence or survival in glioblastoma. These combined tools allow neurosurgeons to make decisions based on data that lead to positive outcomes [53]. AI algorithms are being used to identify new biomarkers and therapeutic targets. For example, the integration of multi-omics using ML has uncovered new molecular pathways that are involved in glioblastoma progression, which could lead new therapies to treat patients in a personalized fashion [54].

Perhaps the most transformational advancement in precision neurosurgery is the implementation of in real-time genomic editing. CRISPR-Cas9 was developed as a laboratory tool and is being explored for its utility in real-time corrections of pathogenic mutations during surgical procedures. For instance, CRISPR strategies targeting EGFRvIII mutations in gliomas will be explored with a potential objective of disrupting oncogenic signaling. In epilepsy, gene editing modalities that target hyperexcitable circuits will be utilized as a means to achieve sustained seizure control [55,56]. Global initiatives, such as The Cancer Genome Atlas (TCGA) and the Human Brain project, are also contributing to incorporating genomic medicine into neurosurgery via data sharing and collaboration on a grand scale [57]. These projects are enabling the creation of molecularly informed neurosurgical strategies, thereby ensuring that the benefits of precision neurosurgery will be available across the globe [58]. CRISPR-Cas9 is a novel and fascinating advancement in precision neurosurgery that affords real-time editing and corrections of pathogenic mutations intraoperatively [59]. As illustrated in Figure 2, CRISPR will identify target gene(s) and achieve a double-stranded break (DSB) using the designed RNA before executing repairs via non-homologous end joining (NHEJ) or homology-directed repair (HDR). CRISPR, in this case, affords the therapeutic editing of pathogenic genetic mutations rather than the traditionally used, almost exclusively surgical approaches that both opened skulls and visualized brain anatomy. This has been further explored in the targeting of oncogenic mutations to disrupt genetic drivers (specifically, EGFRvIII mutations that are characteristic of glioblastomas) related to glioblastoma treatment. This may enter therapy practice, resulting allowing cyst PRISPR scientists to silence hyperexcitable circuits that are targeted by genetic drivers (e.g., SCN1A, SCN8A) related to seizures in the context of epilepsy. This further highlights the versatility of CRISPR.

Advances such as nanoparticle carriers and adeno-associated virus (AAV) vectors are being researched and optimized for neurosurgery, setting out to address the challenges of off-target effects and crossing the BBB [60]. These advances are exciting, providing hope for tumor-specific therapies, as well as addressing the genetic underpinnings of neurodevelopmental and degenerative diseases. Single-cell RNA sequencing (scRNA-seq) is providing more depth of understanding of the cellular complexity of gliomas and other brain tumors [61]. scRNA-seq has the ability to identify sub-populations of cells resistant to therapy, i.e., stem-like cells, to develop new targeted therapies to prevent or eliminate the aggressive and evasive biology of the tumor [62]. For epilepsy, single-cell profiling has facilitated the identification of novel transcriptomic signatures located in the epileptogenic zone, forming the basis of more accurate surgical interventions for non-lesional epilepsy patients [63].

When combined with technologies that provide real-time intra-operative information, these data are of crucial importance for the surgeon, e.g., repositioning information for resection margins or avoiding essential areas of the brain [64]. A key intra-operative decision making technology is DESI mass spectrometry, which allows for the effective molecular characterization of brain tissue [65]. By surveying the distribution of lipids and metabolites, the surgeon will be able, nearly in real-time, to differentiate tumor margins versus regular brain tissue. This is particularly relevant for gliomas, where the surgical goal is to maximize tumor re-resection while maximizing functional brain [66].

In the case of epilepsy, this aligned dynamic will similarly allow for delineating pathological metabolic markers of hyperexcitable states, targeted to assist resection accuracy. The superordinate ability to selectively query and engage information in real-time during surgery greatly reduces the rate of incomplete resections and its concurrent downstream effects, whereby incomplete resections can lead to post-operative complications. Multi-omics, genomics, transcriptomics, proteomics, and metabolomics provide great granularity of the totality of CNS pathologies. For GBM, proteogenomic studies have shown that many genetic mutations with post-translational modifications occur within biological molecular networks, promoting tumor aggressiveness and resistance to therapy [67]. On numerous occasions, this has led to combination therapy design, both upstream, i.e., genetic mutations, and downstream, i.e., protein pathways [68].

For epilepsy, metabolomics is uncovering the presence of aberrancies and abnormalities, forming the basis of the creation of vulnerability in hyperexcitable states, while transcriptomics forms the genetic basis for the noted breaches in the workup of epilepsy patients. Identifying the right care solution is not enough to begin to engage in personalized standardized care [69,70].

Together, this shift toward customized, precision directed neurosurgery has catalyzed systemic genomics models in practice. Next, a summary of the basic principles of genomics will serve as the basis of an initial discussion of the fundamentals. In the following sections, we will discuss how the technologies of sequencing, bioinformatics, and molecular profiling are shaping the future of neurological disease treatment. In the following sections, we will review the history and summarize materials through the lens of genomics and paradigms, as well as discussing how genomics has evolved in the broad landscape of neurosurgical care.

## 3. Genomic Medicine: An Overview

The growing use of genomic medicine in neurosurgery is changing the discipline and supplying novel methodologies, allowing us to understand and intervene in the molecular complexities of neurological disease. This section will address some of the core principles of genomic medicine, some of the new technologies catalyzing the field, and some applications of genomic profiling, especially with respect to the new intersections that genomics introduces into neurosurgical medicine.

### 3.1. Fundamentals of Genomics

The term genomics, broadly speaking, refers to the complete DNA sequence of an organism, including protein-coding genes, non-coding sequences, and regulatory elements. Genomics differs from classical genetics, which restrictively considers single-gene traits; thus, genomics examines relations between multiple genes and the epigenetic regulation of these genes, leading to a multilayered understanding of the course of complex diseases [71,72].

The field of genomic medicine was established with the resolution of the Human Genome Project in 2003, which mapped approximately 20,000 protein-coding genes in the human genome. This project propelled biomedical science forward, allowing for the discovery of disease-causing genetic variants and the application of cutting-edge technologies for diagnostic and targeted treatments [73]. Since then, the field of genomics has opened the door to understanding the genetic disposition to many neurologic diseases, including studying IDH mutations in gliomas and IDH1 mutations in glioblastomas, as well as understanding SCN1A mutations associated with epilepsy [74].

In neurosurgery, genomic information about tumors and diseases in general is particularly impactful, given the complexity of CNS pathogenesis at the molecular level. Most neurological diseases have both germline mutations (inherited variants present in virtually every cell) and somatic mutations (acquired mutations specific to each individual tumor or diseased tissue) [75]. For example, in glioblastoma, EGFR amplification accounts for some of the aggressive behavior typically observed, while germline mutations in LRRK2 increase the likelihood of PD in individuals who carry them [76]. These insights highlight the heterogeneity that exists within CNS and support the consideration of molecular information in the management of CNS.

New areas of genomics research are focused on synthetic lethality and interactions with non-coding RNAs. Synthetic lethality refers to an instance when the inhibition of two genes results in cell death, whereas the inhibition of one of the genes does not induce cell death, acknowledging that other biophysics may play a role in the definition. The principles of synthetic lethality have been used to define the cell dependencies of glioblastoma cells. In these cases, gliomas with IDH mutations may depend on alternate pathways to exist; therefore, targeting could be introduced as a therapeutic action [28,77]. Non-coding RNA, such as microRNAs and lncRNAs are increasingly being identified as biomarkers for diagnoses and targets for therapy, just as they are important regulators of gene expression [78].

Epigenomics extends these principles by questioning how chemical modifications to DNA and histones can affect the expression of genes. One specific example is in glioblastomas, i.e., the methylation of the MGMT promoter silences a DNA repair enzyme which increases the sensitivity of the cells to the alkylating chemotherapeutic. Epigenetic modifications are also inherently dynamic and reversible, meaning that they may be a more optimal pharmacological target [79].

### 3.2. Technological Advances

Translating genomic medicine into practice with neurosurgery is taking place because of innovations in sequencing technologies, bioinformatics, and molecular visualization technologies. Not only do these technologies promote new discoveries but also allow these molecular discoveries to be translated into patient care.

NGS is at the forefront of genomic medicine, bringing about advances in the ability to sequence whole genomes (WGS) and target gene panels, as well as to be used in charged environments. In neurosurgery, NGS can identify driver mutations, mutational signatures, and structural rearrangements in tissue isolated from brain tumors that could be used to characterize drivers to inform treatment plans. For example, the detection of BRAF mutations in pediatric low-grade gliomas using next generation sequencing has promoted the establishment of successful drug development programs to create targeted inhibitors [80,81].

Moreover, single cell sequencing has brought about unprecedented understanding of cellular heterogenicity in neural diseases. For example, in glioblastoma, single cell RNA sequencing has allowed for the identification of variations in tumor cell subpopulations that have distinct transcriptomic profiles and possess treatment-resistant stem-like properties [82]. This level of granularity will provide the basis for therapies that focus on the treatment of specific cellular subtypes rather than treatment of the tumor overall. Similarly, as relates to studies of epilepsy, single cell sequencing has provided insights into the abnormal neuron and glial interactions that drive hyperexcitability, uncovering targets that are amenable to intervention, as they can now be studied in isolation in terms of their true functions [83].

Recently, a third technology, spatial transcriptomics, has emerged; this provides the possibility of mapping gene expression in the spatial context of tissue architecture. Spatial transcriptomics has significant implications in neurosurgery due to the importance of understanding the tumor microenvironment. For example, in gliomas, spatial transcriptomics has provided evidence of immunosuppressive tumor niches, allowing tumor cells to evade the anti-tumor effects of the immune system, thus shaping immunotherapeutic approaches [84].

CRISPR-Cas9 is not only a tool for gene editing, but also a platform for functional genetic exploration. CRISPR screens have been implemented to identify essential genes in glioblastoma and show capacity to identify the vulnerabilities of cells to targeted therapy [85]. For instance, in preclinical studies, the CRISPR based disruption of metabolic pathways in IDH mutant gliomas appeared promising. In addition to gene editing, diagnostics using CRISPR (e.g., SHERLOCK, DETECTR) are in development for the rapid, cost-effective, and easy detection of mutations in clinical samples [86,87].

Long-read sequencing technology (e.g., Pacific Biosciences, Oxford Nanopore) offers a way to resolve complex structural variants and repetitive sequences that are undetectable with short-read technologies. Thus far, it has revealed new mechanisms of chromosomal instability in brain tumors that have yet to be examined. Epigenomics is advancing rapidly, providing novel therapeutic targets in the forms of histone modifications and enhancer elements that drive oncogene expression [88].

### 3.3. Genomic Profiling

Genomic profiling applies sequencing technologies to bioinformatics, creating an extensive molecular characterization of a disease. In neurosurgery, genomic profiling is transforming brain tumor, epilepsy, and neurodegenerative disease management, providing accurate diagnostic, prognostic, and treatment insights [89,90,91].

Genomic profiling has shifted the model of diagnostics by detecting molecularly defined subtypes not detected by histopathology alone. For example, IDH mutations and 1p/19q bi-fold deletions have shifted the collection of diffuse gliomas to the molecular classification level, allowing clinicians to more accurately stratify patients and inform eventual prognosis predictions [92,93]. Similarly, genomic profiling has detected monogenic causes of epilepsy, such as SCN1A mutations in Dravet syndrome, which lead to precise molecular diagnoses and treatment plans [94].

Biomarkers that predict prognoses for disease progression and treatment response are identified through genomic profiling. MGMT promoter methylation in glioblastoma is a reliable predictor of temozolomide sensitivity; meanwhile, mutations in TERT and TP53 indicate more aggressive tumor behavior [9]. Furthermore, genetic mutations for a variety of mTOR pathway genes, such as DEPDC5 in epilepsy, suggest sensitivity to mTOR inhibitors, providing evidence that genomic profiling can play a role in patient treatment selection [95].

Genomic profiling is an important component for targeted therapies, where personalization is distinctively possible. For instance, glioblastomas with EGFR amplification may be sensitive to EGFR inhibitors, and tumors that are BRAF mutant-dependent can be treated with BRAF/MEK inhibitors [96]. Gene-therapy is also emerging as a new paradigm for the correction and replacement of defective genes, with clinical trials on the use of AAV vectors for PD and SMA showing promise [97]. Combining omics with genomics brings a more holistic approach to the study of disease biology. For instance, proteogenomic studies in glioblastomas have revealed post-translational modifications driving therapy resistance, while metabolic profiles demonstrate new vulnerabilities in tumors with IDH mutations. Insights into tumors are paving the way for combination therapies that target multiple levels of tumor biologies at once [98].

In summary, this section has set the stage for a greater understanding of the principles and applications of genomic profiling. Next, we will see applications of these approaches to neurosurgical problems in practice. The next section will see the integration of genomic profiling into practice in neurosurgery to provide evidence of how the technology and principles described here can be applied to improve the care of patients affected by brain tumors, epilepsy, and neurodegenerative disease, as displayed through formal case studies and novel applications in practice.

## 4. Integrating Genomic Insights into Neurosurgical Practices

Using genomic knowledge in neurosurgery is not merely an additional layer of practice; rather, it represents a conceptual shift in terms of how we think about, diagnose, and treat neurological disorders. Genomic medicine reveals the underlying molecular and cellular mechanisms of disease—extending far beyond visible lesions. With genomic profiling technologies that are now capable of decoding the genomic blueprints of tumors, epileptic foci, and neurodegenerative processes, neurosurgeons can incorporate an additional layer of precision into their clinical practice.

### 4.1. Genomic Profiling in Brain Tumors

The molecular characterization of brain tumors has ushered in a new era of diagnostic certainty and therapeutic specificity. Historically, the classification of brain tumors relied heavily on histology and imaging. Today, genomic data complement—or, in many cases, supersede—traditional approaches, providing a deeper understanding of tumor biology and behavior.

#### 4.1.1. Gliomas

Gliomas were among the first brain tumors to demonstrate the benefits of genomic classification. IDH mutations have fundamentally reshaped our understanding of glioma biology. IDH-mutant gliomas produce the oncometabolite 2-hydroxyglutarate, which reprograms cellular metabolism and inhibits differentiation. These biological changes are strongly correlated with prognosis: patients with IDH-mutant gliomas exhibit slower disease progression and improved survival compared to those with IDH wild-type tumors. This prognostic correlation has significantly influenced surgical strategy, with IDH-mutant tumors considered suitable candidates for more aggressive resections due to their well-defined boundaries [99,100].

MGMT promoter methylation, which silences a DNA repair enzyme and increases sensitivity to the chemotherapeutic agent temozolomide, represents another crucial molecular characteristic. By stratifying patients according to MGMT methylation status, clinicians can optimize therapeutic regimens, minimize treatment-related toxicity, and improve outcomes for glioblastoma patients, particularly in cases where resistance is suspected [101]. Single-cell sequencing has further deepened our understanding, revealing glioblastomas to be heterogeneous and dynamic ecosystems rather than homogeneous masses. Within these tumors, therapy-resistant subpopulations, including stem-like cells, drive recurrence and complicate treatment. Ongoing research aims to delineate these subtypes to identify novel therapeutic targets and explore combinatorial approaches designed to eradicate resistant subclones [102].

#### 4.1.2. Meningiomas

Meningiomas, traditionally considered benign, exhibit significant biological diversity at the genomic level, challenging this simplistic classification. Mutations in TRAF7 and KLF4 are typically associated with less aggressive subtypes, whereas NF2 mutations are linked to higher-grade, recurrent tumors. This molecular insight is invaluable for surgical planning, particularly in cases of incomplete resection, where adjuvant therapy may be required.

Epigenomic profiling has further refined meningioma classification. DNA methylation signatures are emerging as powerful prognostic tools, offering insights into tumor grade and recurrence risk [44,103,104]. Such data are critical for guiding surveillance strategies, enabling de-escalation of interventions in low-risk tumors while ensuring close monitoring and timely intervention for aggressive subtypes.

#### 4.1.3. Other Brain Tumors

Genomic profiling has also transformed the classification of less common brain tumors, such as medulloblastomas and ependymomas. Medulloblastomas are now stratified into four molecular subgroups, i.e., WNT, SHH, Group 3, and Group 4, each with distinct prognostic and therapeutic implications. For example, WNT-subtype medulloblastomas respond exceptionally well to therapy and may benefit from treatment de-escalation, whereas Group 3 tumors are associated with poor outcomes and require intensified treatment regimens [105].

In ependymomas, genomic characterization of RELA-C11orf95 fusion genes is refining the diagnostic criteria for aggressive variants and informing clinical trials targeting biologically defined subtypes [106].

### 4.2. Personalized Surgical Planning

The integration of genomic data into surgical workflows enables neurosurgeons to operate with not only anatomical but also molecular precision. Surgical strategies can now be tailored to the patient’s unique genetic and epigenetic profile, improving outcomes while mitigating risk.

In glioblastomas, fluorescence-guided surgery using 5-ALA has revolutionized intraoperative visualization by exploiting the metabolic properties of tumor cells to identify malignant tissue in real time [107]. Genomic factors, such as MGMT methylation status, may further influence 5-ALA uptake and inform surgical planning [108].

Pre-operative genomic mapping combines advanced imaging with molecular profiling to delineate tumor margins more accurately. Tumors harboring IDH mutations or 1p/19q co-deletions are generally more localized and less invasive, allowing for more aggressive yet safer resections. Conversely, glioblastomas with EGFR amplification are typically diffuse and infiltrative, necessitating a delicate balance between maximal resection and functional preservation [8].

In epilepsy surgery, patients with drug-resistant seizures are increasingly undergoing genomic profiling to identify previously undetectable epileptogenic zones. Genes such as SCN1A and DEPDC5 are helping pinpoint these zones when conventional imaging fails [109]. This approach is particularly valuable in non-lesional epilepsy, where genetic insights are guiding the application of minimally invasive techniques such as LITT [110].

### 4.3. Intraoperative Genomic Applications

Operating rooms are rapidly evolving into centers of molecular precision, with real-time genomic tools shaping intraoperative decision-making.

Portable sequencing platforms, such as Oxford Nanopore’s MinION, now enable intraoperative genomic analysis. For instance, the rapid determination of IDH mutation status during glioma surgery allows surgeons to adjust resection strategies based on the tumor’s molecular profile [111]. This capability reduces the risk of incomplete resection while preserving critical brain structures [112].

Intraoperative mass spectrometry (e.g., DESI) provides high-resolution molecular maps of the brain, distinguishing tumor from normal tissue with remarkable accuracy. In glioblastoma surgery, DESI has successfully identified molecular markers of resistance, informing intraoperative strategies and potential adjuvant interventions [113,114].

CRISPR-based diagnostics are surfacing as powerful intraoperative tools to detect specific mutations from tissue samples in real-time. To provide on-the-spot feedback to a surgeon, CRISPR-Cas9 systems can, for example, identify EGFR amplifications and TERT promoter mutations within minutes [115]. These tools should be able to be incorporated seamlessly into the daily workflows of neurosurgical care and ensure that both neurotherapies and surgery are collaborative and aligned with the molecular pathology underlying the target condition [85,116].

The genomic revolution is fundamentally reshaping neurosurgical practice and expanding the possibilities for precision and personalized care. The following section in this paper will extend this discussion to other neurological disorders, including AD, PD, and genetic epilepsies. Precision medicine principles will continue to guide these advances, signaling a future in which genomic neurosurgery may redefine the treatment of some of the most complex and challenging neurological conditions [117,118].

### 4.4. Limitations, Barriers, and Ethical Considerations

While the integration of genomic, transcriptomic, and artificial intelligence technologies into neurosurgery offers unprecedented opportunities to transform diagnostic and therapeutic paradigms, significant challenges remain.

A primary limitation is the real-time integration of multi-omics data streams. Although substantial progress has been made in collecting large-scale datasets, spanning single-cell RNA sequencing, epigenomic mapping, and proteomics, these modalities have yet to be seamlessly combined in the operating theater. The absence of ultra-low latency data pipelines capable of processing high-dimensional data and interfacing with imaging and feedback systems remains a major barrier [119]. Furthermore, dynamic multi-omics models trained on temporally resolved intraoperative data are still in their infancy, and clinical-grade interfaces for real-time data interpretation are lacking [120].

Most institutions and sites cannot currently afford the infrastructure required for these technologies, be they portable nanopore sequencing, mass spectrometry based diagnosis of tissues such as DESI and iKnife, or even surgical robotics utilizing artificial intelligence. For example, the Oxford Nanopore device known as the MinION (Oxford Nanopore Technologies Ltd., Oxford, UK) requires at least one operator, as well as wet lab support and highly sophisticated base-calling and variant-calling software pipelines; in time limited or resource restricted contexts, this is unlikely to be feasible [121]. AI platforms such as DeepMind’s AlphaFold or PathAI are heralding revolutionary advancements in prediction and pattern recognition, against which previous generations of technology cannot compete; however, these technologies can be considered to be in the early stages of clinical translation and will require large annotated datasets that may not exist for rare neurosurgical diseases. Further, licensing, validation, and regulatory pathway approval of these technologies is highly variable among countries and health systems, and the multitude of asynchronous pathways for integrating with clinical care is challenging [122].

The genomicization of neurosurgery creates significant ethical and legal implications. The intraoperative application of patient-specific germline and somatic genomic data raises questions of incidental findings (such as discovering hereditary cancer syndromes), re-consent from operating rooms, and implications for family members sharing pathogenic variants. Additionally, the long-term storage context and the secondary use and potential sharing of these datasets, particularly for machine training purposes, require navigation through complex landscapes of GDPR, HIPAA and other judicial regulations [123]. Additionally, AI applications in neurosurgical decision-making increase explainability and accountability issues. For example, what if an AI model predicts that a certain extent of a resection for a specific tumor will optimize survival based on an analysis of socio-economic data from multiple institutions, but the specific patient’s tumor is biologically different from the normative dataset of the tumor? Who defines the surgical decision rules associated with AI recommendations? Current explainable AI (XAI) methods are grossly insufficient in terms of providing adequate transparency for high-risk clinical decisions [93].

Although the pace of discovery has accelerated, significant gaps remain in terms of bench-side feasibility versus clinical reliability. For instance, CRISPR-based diagnostics (i.e., SHERLOCK, DETECTR) or intraoperative profiling of methylation via nanopore sequencing have demonstrated proof-of-concept efficacy in pilot studies but not longitudinal outcome data in surgical cohorts [124]. Few randomized controlled trials, underpinned by adequate funding, have been undertaken to compare these technologies in either head-to-head comparisons or standard care pathways. Furthermore, the very logistics of embedding molecular diagnostics into the surgical processes, often with timescales measured in mere hours, require detailed pre-operative preparation workflows, intraoperative decision trees, and multidisciplinary clinical support teams, none of which currently exist. Reimbursement models remain ambiguous, particularly for intraoperative molecular diagnostics, and cost-effectiveness has not been demonstrated in diverse populations in any meaningful way [125].

Finally, genomics-enabled neurosurgery is presently concentrated within high-income academic medical centers around the world, which portends a hugely significant disparity of neurologic care risk. Most datasets that underpin the training of AI models are almost exclusively populated by North American and European populations, which leads to questions around the generalizability and unintended biases that could harm underrepresented and identified groups in surgery practice when these training datasets are applied clinically [126]. The implementation of genomic workflows in low- and middle-income countries would require technology not just the to make it affordable but also to develop the workforce, cloud-infrastructure systems, and consistent, multilingual, inclusive, and culturally competent consent processes for patients or their families, which can also not be assumed to be standard [127].

## 5. Genomics and Emerging Neurological Diseases

The genomic revolution is significantly changing how neurological diseases are recognized and treated. From revealing the molecular causes of neurodegeneration to identifying genetic pathways behind epilepsy and rare diseases, genomic medicine provides unique opportunities for precision diagnostic and targeted neurosurgical treatments.

### 5.1. Neurodegenerative Diseases

Neurodegenerative disease is defined as the progressive loss of neurons and subsequent deterioration of their functions. Its management has taken a symptomatic approach in the past. Genomic investigations are untangling the complicated interaction of genetic, epigenetic, and environment factors contributing to these disorders and allowing for treatment strategies to develop and improve.

#### 5.1.1. Alzheimer’s Disease

AD is defined by the associated accumulation of amyloid-beta plaques and tau tangles, and genomics has found much more to consider. Apart from carrying the APOE ε4 allele that increases risk and accelerates disease progression, GWASs have identified common variants in TREM2 which are involved in the regulation of microglial activity and in BIN1, which affects synaptic health. Therapeutic targets are increasing and not only include amyloid-beta, but also include microglial regulation and synaptic preservation [128].

New studies demonstrate that APOE ε4 enhances tau phosphorylation and spread, regardless of amyloid-beta. This is leading to the development of anti-tau monoclonal therapies. CED therapies are currently being researched to deliver drugs directly to the target areas of the brain and bypass the BBB [129]. FUS is also a novel method; it is being utilized to transiently open the BBB while enhancing drug delivery along with stimulating endogenous microglial activity to clear amyloid plaques [130].

Epigenomics is revealing further mechanisms of AD pathology. There is evidence linking the aberrant DNA methylation of genes related to synaptic plasticity and cognitive decline, including BDNF [131]. Histone deacetylase inhibitors (HDACis) are being developed to promote normal gene expression in the earlier phases of disease, when neuronal networks can still be salvaged [132].

#### 5.1.2. Parkinson’s Disease

PD has a heterogeneous genetic underpinning, with mutations in LRRK2, SNCA, and GBA having important roles. LRRK2 mutations result in increased kinase activity with impaired autophagy and lysosomal clearance of alpha-synuclein aggregates. Small-molecule LRRK2 inhibitors are being tested in later phase trials, with preliminary reports demonstrating reductions in neuronal toxicity [133,134].

Mutations in GBA are associated with the most debilitating motor and cognitive symptoms due to defective glucocerebrosidase (GBA) activity. AAV-based gene therapy development is being undertaken to deliver a functional GBA gene to lysosomal compartments in order to restore enzymatic activity and improve the clearance of alpha-synuclein aggregates. Further, transcriptomic and metabolomic advances have identified disrupted bioenergetics and lipid metabolism in mitochondrial processes, leading to the validation of combination therapies targeting several different pathways [135,136].

As these data emerge, DBS strategies are also being re-envisioned. Genetically relevant DBS protocols are being developed with adjustments in both electrode placement and stimulation parameters based on the genetic subtype of each patient. For example, for patients with GBA mutations, targeting circuits that have been implicated in cognitive dysfunction along with motor pathways may provide additional benefit. In parallel, they have also begun to explore dual-intervention therapies with the infusion of gene therapy during DBS implantation surgery, as mentioned previously [137].

Multi-omics approaches are revealing new biomarkers that connect “-omic” (genomic, proteomic, etc.) data to molecular-level dysfunction in neurodegeneration. For example, proteomic investigations have identified hyperphosphorylated tau and truncated alpha-synuclein as important contributors to the progression of AD and PD, respectively. Research on metabolic profiling has discovered mitochondrial dysfunction as a common vulnerability across a range of neurodegenerative conditions and has even suggested a metabolic target for possible neuroprotective strategies [138].

### 5.2. Epilepsy

Epilepsy is a genetically heterogeneous disorder, with more than 1000 genes relevant to the disease spectrum having been identified to date. Genomic profiling is contributing to our understanding of the disease’s basic mechanisms, as well as providing a basis for developing approaches to treatment, particularly with respect to drug-resistant epilepsy. Some examples of monogenic epilepsies that are relevant to the above clinical examples include Dravet syndrome (SCN1A mutations) and tuberous sclerosis (TSC1/TSC2 mutations). SCN1A mutations, which influence the function of the sodium channel and lead to neuronal hyperexcitability and resistance to sodium channel blockers, have changed treatment options, favoring cannabidiol (CBD) or stiripentol, for example, which impact GABAergic pathways [139].

In tuberous sclerosis, mTOR pathway dysregulation leads to seizure activity and cortical tuber development. Thus, everolimus, an mTOR inhibitor, is now a first-line option; it reduces the frequency of seizures and increases the response to surgical resection, such as with cortical tubers. Non-lesional epilepsy has always been problematic for diagnosis and treatment, in that conventional imaging techniques do not demonstrate structural abnormalities. Genomic profiling is beginning to address this problem by identifying mutations to genes, such as DEPDC5 and NPRL3, both of which are components of the GATOR1 complex that modulates both neuronal excitability and metabolic stability. Collectively, this expands the potential options for minimally invasive interventions, such as LITT, according to molecular signatures instead of conventional imaging [98,140].

Single cell sequencing is identifying gene expression signatures related to sites of epileptogenesis and supports intraoperative molecular markers that could potentially increase precision in surgical resections. These discoveries are particularly relevant in complex cases, such as when seizures arise from diffuse or multifocal cortical networks.

### 5.3. Other Neurological Conditions

Genomic breakthroughs are transforming our understanding of rare and poorly understood neurological disorders by generating actionable knowledge that informs new diagnostic and treatment approaches to such conditions. Amyotrophic lateral sclerosis (ALS) exemplifies a condition caused by the progressive loss of motor neurons. Genetic studies have identified causal mutations in genes including C9orf72, SOD1, and TARDBP as central to the pathology of the disease [141]. Hexanucleotide repeat expansions in the C9orf72 gene generate toxic RNA species and toxic dipeptide repeat proteins that compromise cellular homeostasis [142]. Antisense oligonucleotides (ASOs) targeting these toxic RNA species are reportedly entering clinical trials, and ASOs are being tested with intrathecal delivery systems to achieve broad distribution to the spinal cord and brainstem. Additionally, CRISPR-Cas technologies are showing pre-clinical efficacy in excising pathogenic C9orf72 repeats and may offer a cure. Transcriptomic analyses have also mapped the downstream effects of these mutations and will allow for the generation of combination treatments targeting both primary and secondary molecular effects [143].

Hereditary spastic paraplegia (HSP), a group of disorders caused by mutations in genes that regulate axonal transport (e.g., SPAST, ATL1), represents another area of genomic medicine. Mutations cause a loss of normal microtubule dynamics, eventually leading to spasticity and motor weakness. There are efforts to develop microtubule stabilizing agents to reverse axonal dysfunction. There are even gene therapies in development using AAV vectors to deliver wild-type gene copies that could theoretically correct an underlying disease [144]. Recent advancements in cryo-electron microscopy (cryo-EM) have provided structural insights into the spastin–microtubule interaction, which may help in the design of small molecules to restore microtubule severing. Impressive results have also been published concerning combining gene therapy using small shRNA constructs with neuroprotective agents like NAD+ precursors to assess immune enhancing neuronal resilience. Metabolomic profiling has identified enamel vulnerabilities in lipid metabolism and mitochondrial function, paving the way for investigational therapies such as PPAR agonists and mitochondrial antioxidants to slow disease progression [145].

Additionally, spinocerebellar ataxias (SCAs) are an excellent example of the promise of precision medicine for rare diseases, as defined by over 40 subtypes that are predominantly caused by trinucleotide repeat expansions. Mutant genes such as ATXN1, ATXN2, and ATXN3 have toxic gain-of-function mechanisms, resulting in progressive neurodegeneration in the cerebellum and brainstem. There are also advancements in long-read sequencing technologies (i.e., PacBio and Oxford Nanopore) that precisely characterize these repeat expansions to help diagnoses of the subtype of SCA. Given the potential role of RNA toxicity in the pathology of SCAs, ASOs targeting mutant transcripts have been developed, with pre-clinical evaluation in SCA3 mouse models showing a reduction of nuclear RNA foci and motor behaviors [146]. Further, it has been shown that epigenetic dysregulation, mainly aberrant histone acetylation, is also an important contributor for SCAs, with early phase trials of HDACs showing promise in terms of restoring transcriptional activity and improving coordination and motor function. Proteomic studies are further establishing dysregulated calcium signaling and mitochondrial dysfunction as key contributors to neuronal death, allowing for the eventual development of combination therapies aimed at stabilizing mitochondrial function and increasing synaptic plasticity [147].

To therapeutically address these conditions, it is key to overcome challenges related to drug delivery across the BBB. Nanoparticle-based systems are beginning to serve as revolutionary modalities for delivering RNA therapies, such as ASOs and siRNAs, directly to impacted neuronal populations. Functionalized nanoparticles enable cell-specific targeting while decreasing off-target effects. For SCAs, nanoparticles loaded with CRISPR-Cas9 systems are showing promise in preclinical models to precisely excise pathogenic tri-nucleotide expansions. For HSP, nanoparticles containing mitochondrial enhancers and microtubule-stabilizing agents are undergoing testing to restore axonal transport and decrease inflammation. Exosome-mediated delivery systems are similarly demonstrating strong promise for transporting RNA therapies and neuroprotective proteins across the BBB, providing a bio-compatible and highly efficient therapeutic platform [148,149].

CED and FUS are concurrently advancing the precision and efficacy of CNS-targeted therapies. CED is allowing for the uniform distribution of ASOs and other therapeutic agents into defined regions of the brain and spinal cord, as demonstrated by its use and efficacy in decreasing disease markers in SCAs and improving motor function. FUS uses localized microbubbles to transiently open the BBB, which also allows for localized delivery of neuroprotective agents and monoclonal antibodies with minimal whole-body exposure [150,151]. These technological advancements are not only expanding the therapeutic toolkits for rare conditions but also identifying new avenues for the treatment of neurological diseases more broadly [152,153].

The assimilation of genomics as a factor in the study and treatment of rare neurological conditions represents more than a mere advancement; rather, it is a significant paradigm shift. These new tools are affording neurosurgeons the unprecedented ability to address the molecular underpinnings of diseases such as ALS, HSP, and SCAs, giving rise to the possibility of personalized treatment considerations, again based on the molecular characteristics of individual patients. The dissemination of these genomic discoveries into the clinical realm will continue to expand the possibilities of what we can presently do for patients with neurological diseases by applying precision-based neurosurgery principles.

Discoveries (shown in Table 1) using the tools described above are advancing precision in diagnosis and continuing to pave the way for the consideration of rational personalized therapies that can be directed against the specific molecular profiles of each patient.

## 6. Personalized Treatment Strategies

The field of genomics is changing both the future of neurosurgical care and neurosurgical treatment paradigms by enabling treatments that target the molecular and genetic level factors. Targeted therapies, as well gene editing, and immunotherapy, are creating innovative pathways in terms of addressing the etiologies that cause medical issues. Advances in delivery methods, bioinformatics, and the intra-operative utilization of tools are drastically changing approaches to managing neurological diseases. This section of this paper looks further into new scientific advancements in the applicability of actional-patient precision neuroscience surgical care.

### 6.1. Targeted Therapies

Targeted therapies to change expressed molecular pathways, or actional mutations, are becoming part of the neurosurgical tool box. Targeted therapies are an advanced application, allowing for the refinement of treatment to the disease with minimal collateral damage to healthy tissue.

Recently treatment strategies of GBM, a highly invasive and heterogeneous tumor, have applied targeted therapies as a treatment modality. Our previous discussion of actional mutations covered some aspects of the molecular profiling of GBM, including mutations for epidermal growth factor receptor (EGFR) and predicted responses of the molecular profile during patient management. With the described molecular profile of GBM, there have been experimental investigations into inhibiting EGFR, and the rare variant EGFRvIII, through the drugs erlotinib and osimertinib, in the treatment of GBM in addition to radiotherapy, with or without another treatment approach, in order to increase radiosensitivity as well as providing a tumor burden reduction strategy. Trials utilizing these targeted therapies on GBM—like EGFR inhibitors with anti-angiogenesis approaches and the drug bevacizumab, which enhances blood flow to improve drug delivery to the hemoapmies of the tumor, thereby reducing hyperthermic hypoxia— have shown promising results [173].

In addition to GBM, the treatment of IDH-mutant gliomas may also benefit from the use of targeted therapies. Mutant IDH enzymes convert alpha-ketoglutarate (α-KG) to the oncometabolite 2-hydroxyglutarate (2-HG), with phenotypic effects on cellular differentiation cellular processes and the stimulation of tumor development. Vivisidenib and vorasidenib are IDH enzyme inhibitors completing early clinical trials; evidence indicates that the iatrical delivery of these drugs would result in tumor growth reduction, delayed progression of disease after surgery, and improved surgical management. In addition, IDH inhibitors could serve as maintenance therapies to treat residual patients after surgery, i.e., during the transition into post-operative care [100].

In epilepsy, targeted therapies can be used to examine the underlying molecular pathology of seizures; however, the specific molecular mechanisms that drive the generation of seizures are still only partially characterized. This is particularly relevant regarding genetic syndromes such as tuberous sclerosis complex (TSC), where specific mutations in TSC1 and TSC2 cause hyper-activation of the mechanistic target, i.e., the rapamycin (mTOR) pathway. Other genetic forms have dysplastic tissue structures or whole components, meaning that neural connectivities and netburst generate seizures and act as the initial causes of hyperactivity with seizure activity. Everolimus is an mTOR-inhibitor drug that is known to regulate seizure frequency, reduce seizures in the patient with tuberous structures, and improve operational testing (if any) correlating to epilepsy burden and surgical removal [174]. Epilepsy can be said to be nefarious nature, in that we may not see lesions; however, in cases involving the non-lesion category, through genomics profiling, we now have support in terms of surgical management and precision interventions [175,176]. 

Finally, targeted therapies are now starting to be applied in rare cerebral tumor and pediatric brain tumor studies. For example, regarding low-grade gliomas, including those with BRAF gene mutations, we have learned that premalignant or high-grade disease with BRAF mutations tends to have better outcomes through management with BRAF inhibitors. For example, dabrafenib yielded improvements in the symptomatology of a low-grade glioma patient population, and FGFR inhibitors, as the newest approach concerning the kinetics of early clinical low-grade gliomas, has shown possible effectiveness in terms of translation into a clinical practice model [177,178].

Newly emerging technologies or advancements (Table 2) are tackling issues in terms of efficacy and safety.

### 6.2. Gene Editing and Gene Therapies

Gene therapy and gene editing approaches are at the forefront of personalized medicine, as they offer curative capabilities through the correction of genetic defects at their source. Advances in delivery methods, including AAV vectors, coupled with remarkable innovations in genome editing, including CRISPR-Cas9 technology, facilitate the challenges associated with targeting the CNS. CRISPR has been an innovative tool for the live editing of mutations that drive diseases. At the time of writing, researchers have begun to implement CRISPR in the genetic disruption of oncogenic drivers, including EGFRvIII and CDK4 in glioblastomas. Various preclinical studies show significant reduction in tumor invasiveness and improved sensitivity to radiotherapy and chemotherapy. With epilepsy, CRISPR is being used to genetically silence the hyperactive genes (e.g., SCN8A) implicated in severe, intractable seizures [198]. For patients with refractory epilepsy for whom additional treatments are available, these therapies could be routinely implemented into practice [199].

AAV vectors are innovating the delivery of gene therapies by providing stable and long-lasting gene expression in specific regions of the brain. In Parkinson’s disease, AAV-mediated delivery of GBA genes has resulted in restored lysosomal function, decreased toxicity of alpha-synuclein, and slowed neurodegeneration. These therapies may be combined with DBS; therefore, the implementation of two approaches to target both motor and non-motor symptoms of the disease can lead to a synergistic paradigm for improving patient symptoms [200]. In amyotrophic lateral sclerosis, the intrathecal delivery of AAV vectors can genetically silence toxic mutations (e.g., C9orf72 repeat expansions and SOD1 mutations). This method ensures equal gene expression in the spinal cord and brainstem to better address the widespread pathology observed in ALS [201].

Recent advances in delivery strategies may improve the accuracy and safety of gene therapies. CED facilitates the even distribution of therapeutic agents such as AAV vectors or small molecules in targeted areas of the brain while reducing systemic exposure. Intrathecal delivery is proving to be particularly efficacious in the treatment of ALS due to its widespread pathology throughout the CNS. Self-complementary AAVs and cell-type specific promoters are improving gene delivery specificity and efficiency and reducing off-target effects [202].

### 6.3. Immunotherapies

Immunotherapy is an approach that originated in oncology that is being increasingly applied to neurological disorders as a means to utilize the immune system to target a diseased cell while preserving healthy tissue. These therapies are changing the treatment framework for brain tumors and neurodegeneration.

Checkpoint inhibitors, such as anti-PD-1 (pembrolizumab) and anti-CTLA-4 (ipilimumab), are being studied in glioblastoma, a tumor characterized by an immunosuppressive tumor micro-environment. These therapies aim to restore T cell function, which is recognized as a means for the immune microenvironment to recognize and attack tumor cells. Some studies are combining checkpoint inhibitors with radiation therapy, with a resulting synergistic effect in terms of immune activation and tumor control. Post-surgical cancer vaccines that target tumor specific antigens such as EGFRvIII and survivin are being developed as adjuvant therapies to reduce recurrence [203]. Nanoparticles are being studied to deliver or improve the stability, potency, and overall efficacy of these vaccines in terms of improving immune activation while addressing residual tumor cells [204].

In Alzheimer’s disease, immunotherapies are being developed to target microglial pathologies which drive amyloid-beta accumulation and neuroinflammation. TREM2-agonists are being developed to increase the microglial phagocytosis of amyloid plaques, while colony-stimulating factor 1 receptor (CSF1R) inhibitors are being used to enhance the neurotoxic microglial activity. FUS is also being added to these mechanisms to temporarily open the blood–brain barrier to deliver these immunotherapies. In PD, monoclonal antibodies against alpha-synuclein aggregates are being used in concert with genetic therapies to restore lysosomal function, targeting both the molecular and immune pathologies associated with this disease [205].

Emerging immunotherapy innovations include CAR T-cell therapies that engineer T cells to target tumor specific antigens (EGFRvIII). Early phase trials are demonstrating the feasibility of this approach in glioblastoma; CAR-T cells have often evaded traditional immune mechanisms. Studies have explored combining CAR T cells with checkpoint inhibitors to further enhance efficacy, leveraging both innate and adaptive immune responses [206]. Table 3 summarizes key CNS conditions, highlighting causative or associated mutations, the resulting neurofunctional deficits, corresponding genomic or surgical interventions, and the expanding role of artificial intelligence in guiding personalized therapeutic decisions.

## 7. Technological Innovations Enabling Precision Neurosurgery

Neurosurgery is evolving into a precision discipline, a transition that can be attributed in large part to the rapid advance of technologies, such as AI, molecular imaging, bioinformatics, and new tools like optogenetics and nanotechnology. These technologies are driving the translational gap between genetics and the clinic and are offering new frameworks for neurosurgeons to more accurately diagnose, plan, and perform interventions based on personalized treatment that is much safer and more effective.

### 7.1. Artificial Intelligence in Neurosurgery

AI is changing the way neurosurgery is conducted. Investigators are harnessing the power of AI to process, assimilate, and analyze massive and complex datasets, enhancing accuracy in diagnoses and personalizing treatment plans. The strength of AI systems lies in their ability to integrate genomic, radiological, and clinical data to derive robust predictive models that advance neurosurgical decision-making [120,207].

In the field of tumor management, AI-based radiogenomic models are yielding a deeper understanding of the biological behaviors of tumors and resistance to treatment. AI, using ML, integrates imaging biomarkers, such as shape characteristics, texture variations, and enhancement patterns with molecular characteristics, such as IDH mutations and MGMT methylation. In glioblastoma, ML can predict progression and resistance to treatment and can be used in the planning of resections and for the personalization of subsequent treatments. In lower-grade gliomas, AI can identify subtle changes in imaging, i.e., before the clear identification of malignant status, allowing clinicians to intervene in the early stages [221,222].

AI is also disrupting the field of epilepsy surgery. At present, algorithms are being developed to identify informative features from electroencephalogram (EEG) data and are being merged with genomic knowledge to improve accuracy in terms of localizing the foci of seizure activity. This is having the greatest impact in non-lesional epilepsy, where imaging studies were not able to differentiate among epileptogenic zones. AI-enabled robotic systems will also provide surgeons with improved electrode placement capabilities to enhance prediction in stereotactic EEG and DBS, improving accuracy and reducing variability [223]. AI technologies, like Brainlab’s navigation systems, are advancing surgery by integrating real-time imaging with pre-operative data for surgical navigation through challenging anatomies [224].

Another aspect of AI that is transforming the field of neurosurgery is virtual reality and augmented reality (VR/AR). These allow surgeons to perform simulated surgery using patient-specific models created based on genomic, radiological, and anatomical information. More importantly, in constructing a simulation of a difficult patient case using VR/AR techniques, surgeons can enhance their surgical route of approach, leading to a decrease in intra-operative risk, especially in the most sensitive areas of the brain and in deep-seated tumors [225].

Another frontier with transformational implications for neurosurgery is the introduction of AIto live, genomics-directed brain interventions that utilize CRISPR gene editing technology. The use of gene editing to alter the genome in the CNS still has large barriers to overcome—specifically, spatial considerations, the post-mitotic state of neurons, immune barriers, and the irreversible loss of function meaning that the highest possible precision is required, on a molecular level [226]. In this risk averse environment, AI is not simply an analytic engine but a real-time decisions support copilot, utilizing analyses of multi-omics profiles, live electrophysiology, and spatial neuroanatomy to guide genome engineering tools at the neurosurgical interface [227].

In recent months, we have witnessed incredible advances from AI-supported sgRNA design models—such as DeepSpCas9, DeepCpf1, and CRISPR-Net—that have improved the practical clinical use of intraoperative CRISPR. These models utilize deep convolutional neural networks, trained on tens of thousands of cleavage events, to predict on-target efficiency and mitigate off-target mutagenesis across a variable epigenomic landscape [228]. Specifically, these new models consider DNA sequences, chromatin accessibility (as with ATAC-seq), DNA shape, and thermodynamic characteristics to help optimize gRNA selection for the genomic locus in question. In the CNPI, this means editing genes such as SCN1A, DEPDC5, or KCNQ2 in focal epilepsy, or IDH1, TERT, or PTEN in glioblastoma, while also ensuring specificity and limiting collateral damage to the patient [229]. More advanced platforms, such as CRISPRon/CRISPRoff, use even more advanced transformer models to reproduce either the regulatory effect of CRISPR interference or epigenetic modifications, as well as the predictive modeling of how editing noncoding enhancers or promoters could affect the genetic networks of the brain. These AI-based platforms continue to be developed and will also become ubiquitous in surgical workflows. For example, cloud-enabled AI inference engines will offer the ability to optimize gRNAs in a clinically relevant time frame, even intra-operatively in some cases [230]. Some AI algorithms combine integrative multi-omics data, thereby informing intraoperative decision-making concerning molecular signatures and showing the potential to elevate outcomes. Client-based platforms like Seurat, SCANPY, and scVI are allowing physicians to characterize a patient’s tumor single cell RNA sequencing from a high-throughput sequencing workflow and utilize real time data from a clinical workflow [231]. Tools such as Seurat and SCANPY should be noted here. By using existing methods like variational autoencoders and manifold learning, we have been able to determine clusters or populations of transcriptionally unique cells, e.g., invasive glioma stem-like cells or immunosuppressive tumor-associated macrophages (TAMs), with the capacity to model how different cell populations might respond variably to resection, radiation therapy, or gene therapy [232]. For example, using machine learning pipeline-based training on transcriptomic data from spatial transcriptomics could make it possible to determine the subregions of a glioblastoma mass that contains therapy-resistant clones, i.e., with YAP1 activation or hypoxia-inducible signatures, which surgeons could identify with intra-operative imaging [233]. The omitted mapped areas from the omics-handled maps combined with AI-assisted applications for surgical navigation could be added to enable quick and molecularly guided atlases of the brain. Robotic platforms, such as ROSA ONE Brain, Neuromate, and StealthStation S8, could be used to translate this new information to direct viruses, antisense oligonucleotides, or CRISPR payloads to specific subpopulations of cells with heterogeneous lesions [234]. In epilepsy surgery, for example, AI models like EpileptorNet or Spike2Vec utilize the vast quantities of outputs from intracranial EEG (iEEG) with thousands of contact points to identify seizure onset zones via long short-term memory (LSTM) or transformer models. Next, these identified zones could be correlated with known pathogenic variants (e.g., GABRG2, PCDH19) using nanopore sequencing or whole-exome sequencing (WES) to evaluate real-time genomics/electrophysiology fusion for surgical targeting [235].

One of the most ambitious efforts to assimilate AI into this realm is the development of closed-loop neurosurgical systems. These systems sit at the intersection of data from molecular and physiological feedback systems, offering dynamic updates to therapeutic approaches, i.e., in real time, during surgery. In trials, we are currently monitoring real-time CRISPR-Cas9 editing in glioma cells via single-cell RNA-seq and surface proteomics (ex. CD44, Nestin, PD-L1) during intraoperative transitions [236]. AI models can use the previously reconciled temporal -omics changes to learn from spatial outputs to anticipate resistance mutations (e.g., EGFR C797S) or epigenetic escape (e.g., MGMT reactivation) and can suggest adjustments for sgRNA pools or even suggest moving to base editing modalities when they identify evolving potential resistance prompts through reinforcement learning and representative Markov decision processes. This new ability to anticipate outcomes represents, at present, a prototype of future adaptive molecular neurosurgery [227].

In addition, AI systems have begun to be able to model long-term therapeutic effects. Recurrent neural networks (RNNs) or generative adversarial networks (GANs) have been utilized to simulate dynamic neural network interactions, synaptic connectivity, or glial scarring in the months following specific gene edits. These simulations are now being integrated into digital twin networks, from which patient-specific virtual brains can be projected and continually updated in real-time to simulate and predict therapeutic response [237]. AI has emerged as the primary driver of genomic neurosurgery, integrating mechanistic modeling of gene editing, high-resolution molecular state profiling, evolutionally dynamic direct adaptation during therapeutic interventions, and predictive simulation of long-term therapeutic outcomes. In synthesis, the combination of AI and CRISPR with multi-omics is an intelligent surgical approach that is self-adapting, cell-aware, and uniquely individualizable, targeting specific neural microenvironments. These environments are paradigm-shifting surgical spaces, i.e., from static, anatomical based surgery to adaptive, molecularly resolved, neuro-interventions [238].

As AI transforms from a correlative analytics engine to a causal therapeutic architecture, its fusion with neurosurgery enters a new paradigm of uniquely rational, intervention-aware brain therapeutics at the molecular level. In the context of epigenetic dysregulation, now understood to be a key mechanism driving glioblastoma heterogeneity, resistance evolution, and immune escape, AI systems trained on multi-omics dataset atlases such as TCGA, ENCODE, and GTEx can provide latent representations of integrative chromatin states, DNA methylation, noncoding regulatory variants, and enhancer–promoter interactions, as well as employing graph neural networks and transformer-based architectures (e.g., GraphReg, Enformer) to not only identify the dysregulated nodes but also to delineate long-range regulatory circuits, thereby maintaining oncogenic persistence [239]. Once aberrations have been inferred, such as the PRC2-mediated H3K27me3 silencing of differentiation pathways or enhancer hijacking via extrachromosomal DNA elements driving MYC amplification, AI guided platforms can rank intervention strategies through simulated multi-target CRISPR interference, CRISPRa, or programmable epigenetic advisors such as dCas9-TET1 or dCas9-KRAB-MeCP2 [240].

By using deep reinforcement learning architectures trained on longitudinal patient data and high-throughput perturbational datasets (e.g., Perturb-seq, CROP-seq), AI models can now assess the downstream network ramifications of epigenomic editing using cell-type resolved atlases. For example, the AI-guided delineation of epigenetic corrections (e.g., MGMT enhancer re-silencing, CDKN2A reactivation, or reprogramming immunosuppressive tumor-associated macrophages through blockade of a C/EBPβ enhancer) may provide an in silico preview of whether these interventions will translate into the functional restoration of the pathways of interest or will suffer from significant off-target consequences [241]. By leveraging spatial transcriptomic inputs and real-time intraoperative proteogenomic data (most notably, surfaceome drift, as revealed via mass cytometry), closed-loop systems will re-optimize sgRNA design intraoperatively to accommodate any newly detected clonal topographies (such as those that may be implemented through CRISPR-GPS, DeepGuide, or crispr2vec) [242]. Moreover, the dynamic workflows described here transcend static targeting, even when considering IDH1-mutant gliomas with regional hypoxia and HIF1α-augmented YAP1 signaling. AI-augmented systems may also suggest multiple interventions as alternatives, e.g., epigenetic suppression of enhancer looping at TEAD-YAP response elements, in conjunction with viral delivery of synthetic transcriptional repressors to re-establish chromatin equilibrium in resistant subregions [243].

Simultaneously, AI-enabled digital twin platforms (exemplified through NeuroArch, TumorTimeNet) are able to simulate region-specific therapeutic cascades in the brain down to single-cell multi-omics phenotypes, making it possible to test alternative editing strategies in silico prior to a live resection. These simulations will incorporate gene regulatory networks, intracellular signaling dynamics, and estimated resistance evolution due to the intervention pressure they entail, thereby providing a predictive scaffold for neurosurgical decision-making [244]. More ambitiously, generative modeling (including GUI models) using variations autoencoders or GANs provide the capacity for simulating glial scarring, microglial activation, and synaptic rewiring days to months post-editing, effectively creating “therapeutic futures” which can be modeled across a continuum of time and constantly updated based on patient-derived information [245]. It becomes evident that this multi-factorial loop comprising detection, prediction, simulation, intervention, and adaptation is not merely a vision for the future; it is already being utilized in next-generation molecular operating rooms to construct an integrated platform for converting AI (cognition), multi-omics feedback, and gene-editing accuracy into one neurotherapeutic philosophy. This is the rise of closed-loop, epigenetically competent, self-adapting approach to neurosurgery that can respond not just to what the tumor is, but what it will become [246].

### 7.2. Molecular Imaging for Precision Interventions

Molecular imaging is providing neurosurgeons with information regarding molecular and metabolic processes which, in turn, allows precise and dynamic surgical strategies to be devised, such as fluorescence-guided surgery, PET imaging, and intra-operative spectroscopy. Each of these advances will generate exciting opportunities for improved tumor resections and other surgical interventions. Fluorescence-guided surgery, most notably with 5-ALA, has become the standard of care for the treatment of glioblastomas, while assisting surgeons in terms of discerning between tumor tissue and normal brain parenchyma. Newer approaches have been developed, including a next-generation fluorophore which is activated by tumor-specific enzymes (i.e., matrix metalloproteinases (MMPs)) to improve specificity. Genomic profiles can help determine the utility of these fluorophores to achieve precision resections and preserve critical structures in an individual patient [247].

The continued evolution in PET with the initial development of a radiotracer that binds to specific pathways is occurring. For example, [18F]-fluoroethyl-tyrosine (FET) PET, which maps amino acid metabolism in gliomas with grade and aggressiveness, is impacting resection strategies. Likewise, positron emission tomography (PET) tracers that are tau-specific are contributing to our understanding of the pathophysiology of Alzheimer’s disease, offering the ability to diagnose the disease earlier, as well as to customize and target treatments [248].

Intra-operative spectroscopies, including desorption electrospray ionization (DESI) and matrix-assisted laser desorption ionization (MALDI) that evaluate in real-time the molecular characteristics of excised tissue are significantly enhancing the flexibility of margin assessments based on specific biomarkers (i.e., hypoxia signature or DNA methylation status) [249]. DESI has indicated promise for identifying glioblastoma tissue that is likely to show a response to treatment with temozolomide, thus changing the surgical approach and postoperative planning [51,250].

### 7.3. Bioinformatics: Transforming Data into Actionable Insights

Bioinformatics plays a critical role in precision-based neurosurgery, as it assists in the extraction, analysis, and interpretation of data from multiple-omics datasets. Bioinformatics platforms leverage genomic, phospho/proteomic, transcriptomic, and metabolomics datasets to enhance our understanding of disease biology and identify candidate drug targets for the definition of treatment approaches in real-time [82,251].

Bioinformatics are being used to track the clonal evolution of glioblastoma, as well as the timing and mechanisms of resistance. Analytics derived from single-cell sequencing data reveal the heterogeneity of the tumor and inform the design of multi-targeted approaches or combinations of approaches that can account for the subclonal populations. Liquid biopsy platforms that analyze circulating tumor DNA (ctDNA) and extracellular vesicles (EVs) may provide a means to monitor disease response and detect recurrence non-invasively. In epilepsy, bioinformatics have been used to identify gene expression signatures that differentiate epileptogenic tissue from non-epileptogenic tissue; as such, this approach will be used in intra-operative diagnostics to identify seizure-generating zones in real-time. This strategy is especially advantageous for customized resections in challenging cases, i.e., when seizures originate from diffuse or multifocal networks [252].

Neurodegenerative diseases, such as Alzheimer’s disease and Parkinson’s disease, are also benefitting from the advantages of bioinformatics-based approaches that integrate genomic information with cerebrospinal fluid (CSF) biomarkers and structural and functional neuroimaging findings [253]. Predictive algorithms developed using APOE genotyping, tau-PET imaging, and plasma neurofilament light chain (NfL) levels are leading to earlier diagnoses of neurodegenerative diseases and further enhancing the application of targeted therapies. These approaches have also accelerated the creation of precision medicine treatments, including TREM2 agonists for microglial dysfunction and CSF1R inhibitors for neuroinflammation [254].

### 7.4. Emerging Technologies in Neurosurgery

Emerging technologies are expanding the horizon of what we can do in neurosurgery, creating tools that aid in the precision and outcomes of therapies. Optogenetics, or the use of light to activate or inhibit genetically modified neurons, is being investigated as a minimally invasive alternative to traditional surgeries for epilepsy and movement disorders. Within epilepsy, optogenetics is being developed to silence hyperactive neuronal networks without requiring cortical resections. For PD, optogenetic techniques are being used in combination with DBS to optimize stimulation parameters, leading to fewer side effects and better symptom control [255].

Nanotechnology is changing how therapeutic agents can be delivered to the brain. Nanoparticles designed to cross the BBB are being used to deliver chemotherapeutics, gene therapies, or immunotherapies directly to the diseased area. In glioblastomas, nanoparticles containing CRISPR-Cas9 complexes can be used to edit genes directly in the tumor. In Alzheimer’s disease, nanoparticle complexes carrying monoclonal antibodies or anti-inflammatory agents are enhancing the efficacy or specificity of immunotherapy treatments [256].

Robotic tools are a recent technology for increasing the safety and precision of all types of neurosurgical procedures. Robot systems like the ROSA robotic system, that incorporates artificial intelligence and real-time imaging while achieving submillimeter accuracy, are being used for tasks like electrode placement in DBS or biopsies of deep lesions in the brain. These systems are becoming integrated into other minimally invasive procedures, specifically LITT. They increase safety and precision [257].

These technologies are revolutionizing neurosurgical practice. Artificial intelligence, allowing for patient-specific choices to be made in a predictive model, is changing the way we make decisions. New molecular imaging tools also allow for dynamic and real-time guidance intra-operatively. Bioinformatics tools apply multi-omic data synthesis to provide tangible insights, potentially being applied in practice to personalize treatments for tumors and neurodegenerative diseases. Emerging technologies like optogenetics or nanotechnology are pushing the limits of minimally invasive neurosurgery. Robotics are also improving the precision and safety of surgical procedures [258]. Collectively, all these advances are forming a new paradigm in precision neurosurgery, where every decision is based on integrating the molecular characteristics and anatomical features of the patient. The technologies highlighted above will not only provide immediate benefits to neurosurgery practice but will also lay the groundwork for the next leap in future advancements. The next section will consider the impact of these advances in neurosurgical research and practice, specifically looking at the benefits of collaborative, interdisciplinary, centralized data sharing across the globe.

## 8. Future Directions in Precision Neurosurgery

Neurosurgery is being transformed by advances in genomics, multi-omics integration, and new technologies. Advances in genomics are developing into neurosurgical interventions that are increasingly precise, personalized, and effective. Through the advancement of new areas of research, new tools, and international collaboration, neurosurgery is now facing its past challenges.

### 8.1. Expanding the Scope of Genomics in Neurosurgery

Genomics has altered neurosurgical approaches to tumors, epilepsy, and neurodegenerative diseases, but remains an ever advancing field. Whole-genome sequencing (WGS) and scRNA-seq will be able to uncover rare mutations, regulatory elements, and intracellular changes that were previously inaccessible, all of which can provide new targets for therapies.

Recent studies of glioblastomas have also discovered the importance non-coding RNAs, including circular RNAs (circRNAs) and long non-coding RNAs (lncRNAs) with respect to tumor progression and therapeutic resistance. Specifically, circRNAs like circSMARCA5 and lncRNAs like HOTAIR are already under investigation regarding their influence on tumor invasiveness and epigenetic remodeling [259,260]. In addition to understanding the role of non-coding RNAs in tumor behavior, RNA-based therapies are added to the arsenal of therapies under development. In epilepsy, transcriptomic profiling has also advanced our understanding of regional molecular changes in epileptogenic zones. Related to this, single cell analysis has been applied to reveal neuronal subpopulations that uniquely represent regions with dysregulated ion channel expression, as well as to identify regional inflammatory cytokine profiles, leading to therapy development related to those molecular pathways. Due to advances in epigenomic tools like ATAC-seq, researchers are discovering DNA methylation alterations in neuro-degenerative diseases such as Alzheimer’s and Parkinson’s Disease that precede clinical symptoms, providing a potential opportunity for pre-symptomatic intervention with epigenetic therapies [261].

### 8.2. Integrating Multi-Omics Data into Clinical Practice

The integration of multi-omics data—genomics, transcriptomics, proteomics, metabolomics, and epigenomics—will likely revolutionize neurosurgical care by increasing our understanding of disease mechanisms and creating a more personalized treatment approach [262].

Multi-omics approaches are uncovering the synergy between genomic mutations and proteomic and metabolomic alterations that drive the progression of glioblastomas through therapy resistance. For example, proteogenomic studies were able to show that certain post-translational modifications, such as the phosphorylation of critical proteins within the PI3K/AKT/mTOR pathway, contribute to treatment resistance. Further analysis with these proteogenomic studies has provided important insights into the development of combination therapies that target upstream genetic mutations and downstream post-translational modifications to improve therapy response [263].

Research in epilepsy has also used multi-omics to correlate observed metabolic disturbances, such as impairments in glycolysis and oxidative phosphorylation, with the gene expression changes that occur in epileptogenic tissue. This integration is leading to the further development of diagnostic tools that utilize metabolic biomarkers in conjunction with genomic profiling as an approach to improve the localization of seizures and surgical planning. Multi-omics data are also being used to identify novel therapeutic targets, such as the cytokine pathways related to neuroinflammation [264,265].

In neurodegenerative disease, multi-omics data will advance early diagnoses and patient stratification. In the specific case of chemically-induced Alzheimer’s disease, the integration of transcriptomic and proteomic data changes with biomarkers such as phosphorylated tau (p-tau) or NfL is now enabling predictive models concerning clinical outcomes [253]. Predictive models are also being used to specifically identify patients that are most likely to benefit from emerging therapies, such as TREM2-targeting immunotherapies or mitochondrial enhancers [266].

Our understanding of neurological diseases is changing based on multi-omics approaches. These approaches are generating a more thorough understanding (Table 4) of disease mechanisms, revealing potential therapeutic targets of interest, and advancing personalized interventions.

### 8.3. Advancing Technological Infrastructure

Advanced technologies are influencing the precision, safety, and cale of neurosurgical approaches. Newer technologies, such as optogenetics, nanotechnologies, and robotics, are improving the development of therapies whilst reducing procedural risks. Novel imaging and computation platforms are supporting real-time decision making.

Optogenetics is advancing toward clinical application as a minimally invasive alternative therapy for epilepsy and movement disorders. Optogenetics allows light to modulate genetically altered neurons. Within the realm of epilepsy, optogenetics changes the paradigm of resecting cortical tissue. For Parkinson’s disease, optogenetics could be applied with DBS to provide additional control of stimulation parameters for symptom management [284].

Nanotechnologies are facilitating drug delivery across the blood-brain barrier. Nanoparticles have been designed to deliver CRISPR-Cas9 complexes, monoclonal antibodies, and anti-inflammatory agents to diseased areas of the brain. In the context of gliomas, nanoparticles containing chemotherapies and RNA-based therapies will improve specificity and reduce the off-target effects of treatment. In neurodegenerative diseases, nanoparticles as a drug delivery vehicle are increasing the efficacy of therapeutics targeting amyloid clearance and decreasing neuroinflammation pathways [285,286].

Robotics are improving precision and accuracy in neurosurgical procedures. Robotic systems such as the ROSA robotic system have demonstrated submillimeter accuracy in many tasks, including DBS electrode placement and stereotactic biopsies. These systems are currently being used with intraoperative visualization devices, including MRI and ultrasound, to provide real-time feedback during surgery. Robotic platforms with AI are being outfitted with machine learning algorithms that are capable of analyzing intraoperative data to optimize surgical trajectories and minimize injury to surrounding normal tissue [287,288].

### 8.4. Enhancing Global Collaboration and Data Sharing

Global collaboration and data sharing will underpin the evolution of precision neuro-surgery, as it allows the global pooling of resources to occur, confirming findings and contributing to innovation. Large initiatives, including TCGA and the Human Brain Project, will contain repositories of genomic, imaging, and clinical data that will be widely available [289].

Cloud-based platforms and federated learning models can be used for secure data sharing while maintaining fiduciary responsibility. Federated learning permits federating AI algorithms to obtain data from non-centralized databases in order to support models that are robust and generalizable. These processes will be especially valuable and beneficial in addressing disparities in advanced diagnostics and therapies, especially as standardized approaches will promote equity and access to everything that is available globally [290].

Furthermore, for the future support for practicing neural surgeons, we will need standardization regarding the collection and analysis of multi-omics data. International guidelines for such standardization will be the basis of for the next wave of progress in precision neurosurgery, providing benefits for patients, as they will be treated equally, irrespective of their geography or economic means. The development of collaborative networks for research will enhance the scalability of the global progress in such initiatives and will democratize the trial-to-practice process.

### 8.5. Bridging Research and Clinical Practice

Future challenges will involve establishing the applicability of research into clinical practice. Future efforts to be made will have to establish interdisciplinary collaborations to support the transition of new procedures in practice. Adaptive clinical trials that stratify patients based on molecular diagnoses, rather than purely on diagnostic anatomy, are being becoming a standard of practice in the evaluation of treatments in neurosurgery. This approach allows tests of targeted therapies to be carried out in the populations most likely to benefit from them. Examples of this method are basket trials that include patients with the same genetic mutations, including IDH1 or EGFRvIII [291].

In addition, programs that will train neurosurgeons to increase their genomic, bioinformatics, and computational tools literacy, will be necessary. Training programs will help support the next population of neurosurgeons, equipping them with the training and competence they need to better their practice and options for surgery [292].

The future of precision neurosurgery is built on a commitment to continue to investigate personalized and data-driven care. Expanding the applications of genomics to support multi-omics and initiatives to increase cooperation among surgeons will put precision neurosurgery in a position whereby it is able to address neurologic disease paradigms that are not being effectively managed [293,294,295].

## 9. Ethical and Regulatory Challenges in Precision Neurosurgery

As neurosurgery blurs the boundaries of what is possible in medicine, it is also raising important ethical and regulatory questions. These challenges arise from the balance of innovation, equity, and societal responsibilities, and offer productive directions for this exciting field. From genomic data privacy to equity of access to powerful technologies such as gene editing and artificial intelligence, the ways in which we progress with these challenges will impact whether expert neurosurgery can be experienced by all. This chapter will address the ethical questions and regulatory environments that we need to consider in order to support the advancement of neurosurgery.

### 9.1. Ethical Considerations in Genomic and Multi-Omics Integration

The burgeoning use of genomic and multi-omic data in neurosurgery raises substantial ethical questions. On the one hand, genomic and multi-omic datasets provide unparalleled data about an individual’s health, mechanisms of disease, and implications for treatment. On the other hand, these data are intensely personal and can be used to reveal genetic predispositions, risks to family members, and potentially incidental findings unrelated to the purpose of treatment. One of the most profound ethical dilemmas will be: How do we protect patient autonomy while also protecting patient privacy in the shift to the big data era? Genomic and multi-omics datasets (even in a treatment context) come with a real potential for misuse, discrimination, and stigma if proper storage and/or protection of the data is not applied. For instance, a genomic profile may implicate genetic conditions, such as Huntington’s disease or hereditary cancers, for a patient; this could then affect their insurability or employment opportunities. Patient protections will be essential for ensuring that genomic data will only be used for medical purposes. 

Informed consent is evolving to interface with these technologies. Patients need to do more than just agree to the use of their genomic data; they also need to be aware of the implications of incidental findings. For example, if a patient is undergoing surgery for epilepsy and it is discovered that their genome contains a mutation that increases their risk of developing Alzheimer’s disease decades later, is that something the patient should become aware of? When there are no actionable treatments available, how should that information be handled? We need direct conversations that allow patients to make decisions about the extent to which genomic analyses and data sharing can take place.

At the societal level, weighing the overall benefits of shared genomic data against the privacy concerns of the individual is a difficult process. Federated learning models, where algorithms are trained on a decentralized dataset without personal information being shown could be a solution to this challenge. However, trust in the system and an understanding of how the model is transparent will be needed for widespread acceptance.

### 9.2. Equity and Access to Precision Neurosurgery

Precision neurosurgery has the power to revolutionize care, but we do not yet enjoy its benefits equitably. The expense of advanced genomic testing and early, personalized therapies or devices, like robotic surgery, are barriers for low-income patients and underserved areas. If society does not mobilize to mitigate the gap in terms of who can experience the benefits of precision medicine, it will remain a small subset of the population that benefits from it.

Global groups, such as the Global Alliance for Genomics and Health (GA4GH), are striving to democratize access by creating measurable and standardized frameworks and open data sharing models, which return the cost burden to industry. At the same time, the creation and utilization of low-cost sequencing technologies and portable diagnostic technologies will be critical to achieving precision neurosurgery in resource poor environments. Regulatory initiatives also need to foster diversity and inclusion in clinical trials for neurosurgical procedures. Historically, minority populations have not been adequately represented in clinical trials, leading to therapies that may not perform equally well amongst people of diverse genetic backgrounds. Broadening participation helps assure that precision tools and therapies are effective for all populations.

In addition, novel funding models need to be developed to subsidize cutting edge therapies for lower income communities. Governments, non-profits, and private organizations can work together to create programs that facilitate access to technologies like CRISPR-based therapies and advanced imaging technologies in public health settings to help ensure that no one is left behind.

### 9.3. Ethical Implications of Genetic Editing

Gene editing technologies such as CRISPR-Cas9 are some of the most powerful tools ever developed to correct disease causative mutations at their source. While this potential is unprecedented, it also raises a complex ethical questions related to, for example, hereditary genetic changes. Somatic editing, which is when genetic targeting only affects diseased tissues and does not involve germline cells, is the focus in neurosurgery at the present time. Examples of somatic editing include editing SCN1A mutations in Dravet syndrome or targeting EGFRvIII mutations in glioblastomas—both of which are highly promising therapeutic candidates. Somatic editing seems to be an ethically acceptable procedure; it directly benefits the patient and does not involve hereditary changes to the genome that could affect future generations.

Germline editing raises significantly more complex issues. Changing DNA in ways that have an impact on future generations generates significant concerns related to possible unintended and unforeseen effects over the long-term and the potential creation of unequal circumstances. For example, if an individual or groups of individuals were offered the opportunity for genetic enhancement, it is likely these enhancements would increase inequity between those who received genetic enhancements and those who did not, ultimately leading to a “genetic divide.” The development of international agreements and oversight needs to happen as a priority to govern the use of germline editing technologies. There is also an ethical dimension regarding gene editing technologies and their potential use for non-therapeutic use (e.g., cognitive or physical enhancement). Future considerations for striking a balance between the creation of new technologies and preventing misuse will require conversations among scientists, ethicists, policymakers, and the public.

### 9.4. Regulatory Challenges in Precision Neurosurgery

Innovations and advancements in precision neurosurgery seem to move at a pace that is consistently ahead of the ability of regulatory frameworks to respond or adapt. Ensuring that new therapies and technologies are safe, effective, and being used ethically becomes a priority, and it is critical to devise forward-looking regulations that can adapt to the science as it evolves. Gene therapies are examples of technologies that present unique regulatory issues. Delivery systems are becoming more precise, such as nanoparticle and viral vectors; however, important considerations about long-term safety and off-target effects remain. Regulatory agencies such as the FDA and the EMA are working towards developing extensive guidelines for gene therapies that balance vigorous testing and the urgency of providing life-altering therapies and treatment options to patients. AI tools for neurosurgery also bring about regulatory issues that need to be considered. Unlike most traditional medical devices, AI-based systems are dynamic tools that learn and are continuously being updated as new algorithms become available. Regulatory bodies need to create frameworks that encourage adaptability while ensuring uniformity and safety for patients. Established standards for transparency and accountability, and requirements for algorithm validation, must be developed.

Another priority is global convergence regarding data-sharing policies. In order to collaborate effectively across borders, there needs to be formalized protocols for how genomic, imaging, and clinical data are managed and utilized. Federated learning models and cloud-based approaches must be employed in a manner that adheres to ethical and regulatory standards that respect patient privacy while improving international participation in research.

### 9.5. Building Public Trust and Ethical Literacy

For precision neurosurgery to achieve its full potential, public trust in advances in genomics and technology will be required. This will necessitate transparent methods of communicating the benefits and risks, as well as the limitations, of these advances. Healthcare providers and patients will also need to deepen their ethical literacy. There will need to be educational programs for clinicians and the public. Neurosurgeons and genetic counselors will require education to effectively manage complex ethical scenarios and communicate with patients about the consequences of the use of genomic and contextual technologies. Medical education can also be utilized to advocate for patient education campaigns, e.g., for common inquiries regarding the privacy of genomic data and the risks of gene editing being misused. Engaging with patients and stakeholders in the development of ethical guidelines ensures a democratic process of engagement that considers diverse perspectives and practices to promote trust and congruence with societal norms regarding patient autonomy and responsibility. Forums and moderated dialogues may enhance democratic conversations regarding the ethical challenges of precision neurosurgery.

The ethical and regulatory considerations of precision neurosurgery practice represent a complex but a favorable space. By addressing concerns over privacy, equity, genetic editing, and public trust, this field can flourish and broaden its inclusivity. Solutions to the problems cited above exist through possible collaborations, a patient-centered focus of care, and propriety to innovation being commensurate with societal morals. Precision neurosurgery can transform human life globally if an ethical framework that weighs the consequences of innovation is devised.

## Figures and Tables

**Figure 1 ijms-26-07364-f001:**
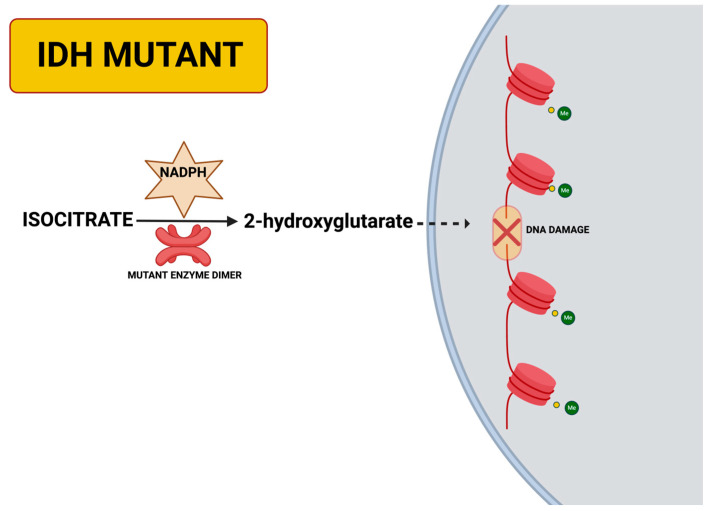
The role of IDH mutations in gliomas. Mutant IDH enzymes convert isocitrate to 2-hydroxyglutarate, an oncometabolite that disrupts cellular epigenetics and promotes DNA damage.

**Figure 2 ijms-26-07364-f002:**
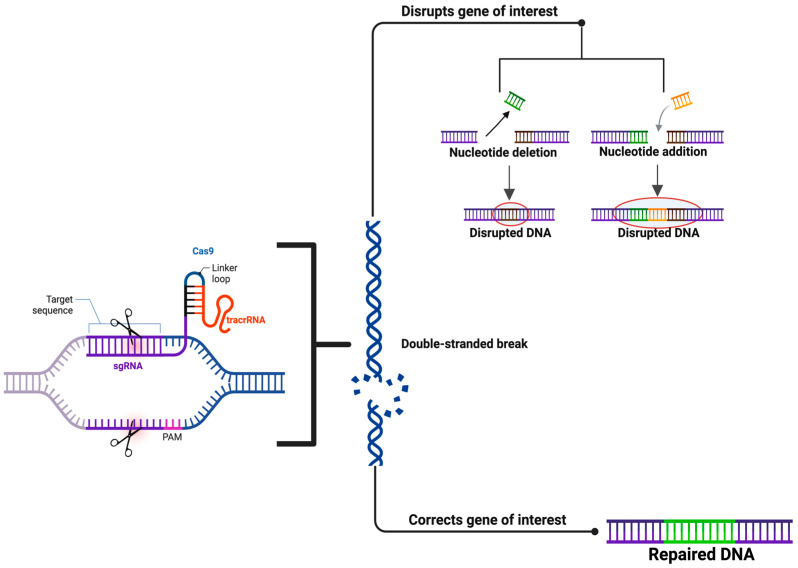
The mechanism of CRISPR-Cas9-mediated gene editing.

**Table 1 ijms-26-07364-t001:** Summary of key mutations identified in various neurological diseases, i.e., glioblastomas, epilepsy, ALS, hereditary spastic paraplegia (HSP), and spinocerebellar ataxias (SCAs). For each condition, the table details the associated mutations, their pathological mechanisms, current therapeutic strategies, advancements in preclinical and clinical studies, and the challenges that remain.

References	Disease	Key Mutations	Pathological Mechanism	Therapeutic Focus	Preclinical/Clinical Advancements	Challenges
[154,155,156,157]	Glioblastoma	EGFRvIII, IDH1, TERT	Tumor proliferation, disrupted metabolism	EGFR inhibitors, IDH inhibitors, CRISPR-based editing	Nanoparticle-based CRISPR delivery under development	Resistance due to tumor heterogeneity
[158,159,160]	Epilepsy	SCN1A, SCN8A, DEPDC5	Ion channel dysregulation, mTOR pathway activation	Cannabidiol (CBD), mTOR inhibitors	Gene-silencing therapies showing reduction in seizures	Incomplete seizure localization in non-lesional epilepsy
[161,162,163]	ALS	C9orf72, SOD1, TARDBP	RNA toxicity, dipeptide protein accumulation	ASOs targeting toxic RNA, CRISPR excision	ASOs delivering uniform therapeutic effects via intrathecal routes	Long-term durability of CRISPR treatments
[164,165,166]	HSP	SPAST, ATL1	Impaired microtubule dynamics	Microtubule stabilizers, gene therapy	Structural modeling advancing small molecule design	Poor BBB penetration of therapeutic molecules
[167,168,169]	SCAs	ATXN1, ATXN2, ATXN3	Trinucleotide repeat expansions	Antisense oligonucleotides, HDAC inhibitors	Positive early-phase trials for HDAC inhibitors improving motor coordination	Need for specific delivery technologies
[170,171,172]	Alzheimer’s Disease	APOE ε4, TREM2, PSEN1	Amyloid-beta aggregation, tau hyperphosphorylation	Anti-amyloid antibodies, tau kinase inhibitors	Microglial-modulating therapies showing plaque clearance	BBB penetration remains limited

**Table 2 ijms-26-07364-t002:** Overview of cutting-edge delivery technologies being developed for CNS disorders, i.e., nanoparticles, exosome-based systems, convection-enhanced delivery (CED), focused ultrasound (FUS), and lipid nanoparticles (LNPs).

References	Technology	Mechanism	Disease Applications	Key Findings	Advantages	Limitations
[179,180,181]	Nanoparticles	BBB penetration, receptor-targeted delivery	Glioblastoma, SCAs	Enhanced RNA delivery to cerebellum; improved survival in preclinical models	Highly specific delivery, reduced systemic effects	Complex manufacturing processes
[182,183,184,185]	Exosome-based therapy	Natural vesicle-mediated RNA/protein transport	ALS, SCAs	High efficiency in delivering ASOs to motor neurons and cerebellar neurons	Biocompatibility, immune evasion	Limited scalability and yield
[186,187,188]	Convection-Enhanced Delivery (CED)	Uniform regional drug distribution	SCAs, glioblastoma	Effective ASO delivery to cerebellum; improved motor outcomes in animal models	Avoids systemic exposure, bypasses BBB	Requires precise surgical techniques
[189,190,191]	Focused Ultrasound (FUS)	Transient BBB disruption using microbubbles	ALS, glioblastoma	Improved localized delivery of monoclonal antibodies and neuroprotective agents	Non-invasive, high localization	Risk of local tissue damage
[192,193,194]	Nanoparticles for CRISPR	BBB-penetrating CRISPR-carrying particles	Glioblastoma, HSP	Demonstrated precise gene editing and reduced tumor growth in preclinical trials	Enables direct gene editing	Long-term safety of CRISPR delivery remains unclear
[195,196,197]	Lipid Nanoparticles (LNPs)	mRNA delivery	Alzheimer’s Disease, Parkinson’s Disease	Effective mRNA transport targeting microglia and neurons	Cost-effective, scalable	Risk of off-target effects

**Table 3 ijms-26-07364-t003:** Selected central nervous system disorders and their relation to commonly implicated genetic mutations, associated functional impairments, and therapeutic strategies under clinical or experimental exploration.

References	Disease	Causative/Associated Genes	Functional Deficit or Pathological Defect	Surgical/Genomic Intervention Strategy	Role of AI
[207,208,209]	Glioblastoma	IDH1, EGFRvIII, TERT, TP53	Uncontrolled proliferation, necrosis, metabolic rewiring	Tumor resection with intraoperative mapping; CRISPR knockout of EGFRvIII; IDH1-specific metabolic inhibition	AI-powered radiogenomics and subtype prediction (e.g., DeepGlioma)
[210,211,212]	Epilepsy (non-lesional/refractory)	SCN1A, DEPDC5, PCDH19, KCNQ2	Hyperexcitability, interneuron dysfunction, mTOR hyperactivation	Stereo-EEG guided laser ablation (LITT); CRISPR correction of SCN1A; responsive neurostimulation	AI fusion of iEEG + variant data for seizure zone prediction (e.g., EpileptorNet)
[213,214]	Parkinson’s Disease	LRRK2, SNCA, GBA	Dopaminergic neuron degeneration, alpha-synuclein aggregation	Deep Brain Stimulation individualized by genotype; targeted SNCA knockdown	AI-based connectome mapping for electrode planning and symptom modeling
[215,216]	Alzheimer’s Disease	APOE ε4, TREM2, PSEN1, BIN1	Amyloid aggregation, tau propagation, synaptic failure	Focused ultrasound delivery of antibodies; future CRISPR APOE allele replacement	AI integration of PET/MRI and genomics for staging and target selection
[217,218]	ALS	C9orf72, SOD1, TARDBP	Motor neuron degeneration, RNA toxicity, protein aggregates	Intrathecal delivery of ASOs or CRISPR constructs; exon skipping or repeat excision	AI stratification of progression trajectories and molecular timing of intervention
[219,220]	Cortical Dysplasia/mTORopathies	MTOR, TSC1, TSC2, RHEB	Focal cortical thickening, epileptogenesis	Surgical resection or ablation of mTOR-active cortex; gene therapy suppression	AI segmentation of cortical lesions using MRI + genotype overlays

**Table 4 ijms-26-07364-t004:** Key findings from multi-omics studies on neurological diseases, spanning glioblastomas, epilepsy, ALS, SCAs, HSP, and Alzheimer’s disease. The table details the omics approaches utilized, the insights gained into disease mechanisms, the clinical implications of these discoveries, emerging therapeutic strategies, and the challenges in translating these insights into practice.

Study/Author (Year)	Disease	Omics Approach	Key Insights	Clinical Implications	Novel Therapies	Challenges
[267,268,269]	Glioblastoma	Proteomics, transcriptomics	PI3K/AKT/mTOR pathway alterations	Guided development of combination therapies targeting genomic and protein-level vulnerabilities	mTOR inhibitors combined with EGFR inhibitors	Resistance due to clonal heterogeneity
[264,270,271]	Epilepsy	Metabolomics, transcriptomics	Glycolysis and mitochondrial dysfunction in seizure zones	Development of metabolic modulators for seizure control	Anti-inflammatory metabolic modulators	Complex metabolic pathways
[272,273,274]	ALS	Transcriptomics, epigenomics	Downstream effects of C9orf72 RNA toxicity	Informed design of ASO therapies addressing both primary and secondary mechanisms	RNA toxicity-focused ASOs	Long-term monitoring required
[275,276,277]	SCAs	Transcriptomics, epigenomics	Histone hypoacetylation, mitochondrial dysfunction	Supported clinical trials of HDAC inhibitors and mitochondrial enhancers	Combination HDAC and mitochondrial therapies	Challenges in targeted delivery
[278,279,280]	HSP	Metabolomics	Lipid metabolism alterations in corticospinal neurons	Targeted use of PPAR agonists and NAD+ precursors for neuroprotection	Microtubule-stabilizing compounds	Difficulties in axonal-targeted delivery
[281,282,283]	Alzheimer’s Disease	Proteomics, metabolomics	Neuroinflammation and tau propagation	Anti-inflammatory and tau-focused therapies improving cognitive outcomes	Tau kinase inhibitors combined with microglial agonists	Lack of early biomarkers for intervention

## Data Availability

The data presented in this study are available on request from the corresponding author.

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
