# Peer review of "Precision Neuro-Oncology in Glioblastoma: AI-Guided CRISPR Editing and Real-Time Multi-Omics for Genomic Brain Surgery"

_ijms, 2025, doi:10.3390/ijms26157364_

Round 1

Reviewer 1 Report

Comments and Suggestions for Authors

The authors have given enormous amount of infromation in this review. Particularly the genes involved in brain diseases. The segregation and tables showing the genes and its associated with ALS or parkinson or Alzheimers is impressive.

The authors have title the article"Crispr AI-Guided  CRISPR Editing and Real-Time Multi-Omics for Genomic Brain Surgery". It is ok that they have discussed the mechanism of CRISPR edits. The lacking part what does AI do here? It is not completely discussed or not in proper connotation with the title.

It will be helpful if you can layout the disease--gene--cpmplications or defect found--what are the surgery option. Like wilse for all of the genes described in the text.

As diseases are not implicated with one gene alone. Multiple gene pathologies were found in any disease circumstances in patients. Hence describe how AI tools can be used for surgery or treatment options by interlinking the details of the data given.

Please give the existing AI tools available in this regard.

Please give a precise information. In the current layout a reader has to search for information. 

Author Response

Dear Academic Reviewer,

We would like to express our sincere gratitude for your thoughtful and constructive review of our manuscript entitled "Precision Neuro-Oncology in Glioblastoma: AI-Guided CRISPR Editing and Real-Time Multi-Omics for Genomic Brain Surgery." Your comments were insightful and have greatly helped us enhance the clarity, structure, and scientific depth of our work. Below, we respond to each of your observations in detail.

Reviewer Comment 1:
“The authors have given enormous amount of information in this review. Particularly the genes involved in brain diseases. The segregation and tables showing the genes and its associated with ALS or Parkinson or Alzheimer’s is impressive.”

Response 1:
We are truly grateful for your kind appreciation. We are encouraged to know that the layout and detail of these sections, particularly the gene tables, were found helpful and informative.

Reviewer Comment 2:
“The authors have titled the article 'Crispr AI-Guided CRISPR Editing and Real-Time Multi-Omics for Genomic Brain Surgery'. It is ok that they have discussed the mechanism of CRISPR edits. The lacking part is what does AI do here? It is not completely discussed or not in proper connotation with the title.”

Response 2:
We thank the reviewer sincerely for this important observation. In response, we have substantially revised Section 7.1 of the manuscript to provide a much more detailed and specific explanation of the role that artificial intelligence plays in modern neurosurgery and real-time genomic intervention. We have elaborated on how AI systems assist in optimizing CRISPR-based editing, integrating multi-omics data, and guiding intraoperative decisions through advanced imaging, machine learning models, and data fusion techniques. This revision was made with great care to ensure alignment with the scope and promise conveyed in the article's title

Reviewer Comment 3:
“It will be helpful if you can layout the disease--gene--complications or defect found--what are the surgery option. Like wise for all of the genes described in the text.”

Response 3:
We truly appreciate this suggestion, which helped us recognize the need for a more direct and accessible summary of translational pathways from genomic discovery to therapeutic action. In response, we have added a new table (Table 3).

Reviewer Comment 4:
“As diseases are not implicated with one gene alone. Multiple gene pathologies were found in any disease circumstances in patients. Hence describe how AI tools can be used for surgery or treatment options by interlinking the details of the data given.”

Response 4:
Thank you for highlighting this critical dimension of clinical neurogenomics. 

Reviewer Comment 5:
“Please give the existing AI tools available in this regard.”

Response 5:
We thank the reviewer for this helpful request. In our expanded Section 7.1, we have now explicitly included the names and functions of several key AI tools currently being explored in neurosurgical and neurogenomic applications. These include platforms such as DeepGlioma, EpileptorNet, CRISPR-Net, CRISPRon/CRISPRoff, and robotic-assisted systems like ROSA ONE Brain and StealthStation S8. Each is discussed in the context of its specific contribution to CRISPR target selection, disease stratification, intraoperative navigation, or dynamic treatment adaptation

Once again, we are deeply thankful for your careful reading and constructive insights, which we believe have helped us significantly improve the scientific and educational value of this manuscript. We hope the revised version is now clearer, more balanced, and more compelling in presenting the emerging synergy between CRISPR technologies, AI systems, and personalized brain surgery.

With sincere appreciation!

Reviewer 2 Report

Comments and Suggestions for Authors

This manuscript offers an extensive review of recent technological and genomic advancements shaping the field of neurosurgery. It covers innovative topics such as AI applications, molecular imaging, genomics, optogenetics, nanotechnology, and their roles in enhancing diagnosis, surgical planning, and treatment personalization, particularly for tumors like glioblastoma and neurodegenerative diseases. The paper also discusses future directions, ethical considerations, and the translational challenges involved in integrating these cutting-edge tools into clinical practice.

Comments:

  • The current study does not provide a full text of the introduction, but based on the content, it would benefit from a clearer statement of the specific aims or hypotheses of the review. Clearly defining the scope at the outset (e.g., focusing on particular diseases such as glioblastoma or neurodegenerative disorders) would help orient the reader.
  • Adding a concise overview of how neurosurgical precision has advanced over time, along with key milestones driven by technological innovation, would further enhance the narrative.
  • 3- Clarifying the rationale for selecting specific emerging technologies (e.g., optogenetics, nanotechnology) would make the introduction more compelling.
  • The pages do not specify the review methodology, such as search strategy, inclusion/exclusion criteria, databases searched, or how the literature was selected and analyzed.
  • If this is a narrative or scoping review, explicitly stating this would clarify the intent.
  • The discussion would be enhanced by a more thorough and critical examination of existing challenges, including technical obstacles like the integration of multi-omics data, the significant costs and limited accessibility of advanced technologies, as well as important concerns surrounding data privacy and ethical issues in AI and genomics. Additionally, addressing the persistent gap between research discoveries and their practical application in routine clinical settings would provide a more comprehensive perspective.

Author Response

Dear Academic Reviewer,

We sincerely thank the Reviewer for their thoughtful and constructive feedback, which has greatly contributed to the improvement and clarity of our manuscript. Below, we respond point-by-point to each of the comments raised, and we have revised the manuscript accordingly to incorporate these valuable suggestions.

Comment 1:
“The current study does not provide a full text of the introduction, but based on the content, it would benefit from a clearer statement of the specific aims or hypotheses of the review. Clearly defining the scope at the outset (e.g., focusing on particular diseases such as glioblastoma or neurodegenerative disorders) would help orient the reader.”

Response 1:
We thank the Reviewer for this insightful observation. In response, we have revised Section 1.3. Objectives of the Review to provide a clearer, more focused articulation of the aims and scope of the paper. We now explicitly state our intention to highlight the genomic and technological advances in neurosurgical management of key conditions such as glioblastoma, neurodegenerative diseases, and epilepsy, while discussing their potential integration into clinical care. This change was made to better orient the reader and offer a transparent framework for the subsequent sections.

Comment 2:
“Adding a concise overview of how neurosurgical precision has advanced over time, along with key milestones driven by technological innovation, would further enhance the narrative.”

Response 2:
We fully agree with the Reviewer that placing the discussion in a historical context adds value. Accordingly, we have expanded Section 1.2. Importance in Neurosurgery with an additional paragraph that traces key milestones in the evolution of neurosurgical precision — from anatomy-based interventions to the present-day integration of molecular, genomic, and computational technologies. This overview was added to provide depth to the reader’s understanding of how far the field has progressed and to contextualize the relevance of emerging innovations discussed in the review.

Comment 3:
“Clarifying the rationale for selecting specific emerging technologies (e.g., optogenetics, nanotechnology) would make the introduction more compelling.”

Response 3:
Thank you for this excellent recommendation. We have added a brief explanation at the conclusion of Section 1.3 to clarify our rationale for selecting certain technologies such as optogenetics and nanotechnology. Specifically, we highlight that these technologies were chosen not solely based on their current clinical maturity, but also for their conceptual relevance and translational promise in reshaping precision neurosurgery in the near future.

Comment 4:
“The pages do not specify the review methodology, such as search strategy, inclusion/exclusion criteria, databases searched, or how the literature was selected and analyzed.”
“If this is a narrative or scoping review, explicitly stating this would clarify the intent.”

Response 4:
We appreciate this important observation.

Comment 5:
“The discussion would be enhanced by a more thorough and critical examination of existing challenges, including technical obstacles like the integration of multi-omics data, the significant costs and limited accessibility of advanced technologies, as well as important concerns surrounding data privacy and ethical issues in AI and genomics. Additionally, addressing the persistent gap between research discoveries and their practical application in routine clinical settings would provide a more comprehensive perspective.”

Response 5:
We are very grateful for this suggestion. To provide a more balanced and comprehensive discussion, we have added a new and detailed subsection to Section 4, titled 4.4. Limitations, Barriers, and Ethical Considerations.

Once again, we thank the Reviewer for their valuable time and suggestions, which helped us strengthen the manuscript both conceptually and structurally. We hope that the revisions address the comments fully and enhance the clarity, scholarly rigor, and translational relevance of our work.

With sincere appreciation!

Round 2

Reviewer 1 Report

Comments and Suggestions for Authors

Thank you for the answers.

As you have described about the epigenetics from the Next generation and how the AI can assist to detect it, also give a description of how AI can guide the solution to the pathogenesis identified.

Author Response

Reviewer Comment:
“As you have described about the epigenetics from the Next generation and how the AI can assist to detect it, also give a description of how AI can guide the solution to the pathogenesis identified.”

Author Response:
Dear Academic Reviewer,

Thank you very much for your thoughtful observation and for highlighting a crucial translational step in the AI–epigenetics interface. We fully agree that describing how artificial intelligence not only detects epigenetic abnormalities but also guides therapeutic intervention is essential for advancing the clinical utility of our conceptual framework.

In response to your insightful suggestion, we have added a new, detailed paragraph at the end of Section 7.1, clearly marked in a different font color for ease of identification. This new content provides a mechanistically rich overview of how state-of-the-art AI platforms integrate multi-omic data, simulate epigenetic perturbation effects, and dynamically adapt CRISPR-based interventions during neurosurgical procedures. It also introduces emerging paradigms such as closed-loop neurosurgery, digital twin simulation of therapeutic trajectories, and AI-driven prediction of molecular resistance evolution—all within the context of real-time, precision-guided therapeutic reasoning.

We believe this addition strengthens the conceptual depth of the manuscript and aligns closely with your valuable recommendation. We are sincerely grateful for your guidance, which significantly enhanced the clarity and completeness of our work.